# Pine Looper *Bupalus piniaria* (L.) Outbreaks Reconstruction: A Case Study for Southern Siberia

**DOI:** 10.3390/insects12020090

**Published:** 2021-01-21

**Authors:** Denis A. Demidko, Olga V. Trefilova, Sergey S. Kulakov, Pavel V. Mikhaylov

**Affiliations:** 1Laboratory of Forest Zoology, Sukachev Institute of Forest, Siberian Branch, Russian Academy of Science, 50, Bil. 28, Akademgorodok, 660036 Krasnoyarsk, Russia; 2Reshetnev Siberian State University of Science and Technology, Krasnoyarskii Rabochii Ave. 31, 660037 Krasnoyarsk, Russia; ovtrefilova_ilsoran@mail.ru (O.V.T.); hbz_sibstu@mail.ru (S.S.K.)

**Keywords:** *Bupalus piniaria*, outbreaks, reconstruction, dendrochronology, seasonwood

## Abstract

**Simple Summary:**

The pine looper damages Scots pine forests over vast areas. However, the study of its population fluctuations is hampered by the lack of long-term observation series. The dendrochronological reconstruction is often used to study the history of its outbreaks. In some cases, such reconstructions require methods that work without comparison with other tree species. We have proposed such a technique based on the analysis of the early- and latewood growth. The technique makes it possible to separate the effect of defoliation on tree rings from weather influence. Besides, it is more sensitive than previously developed methods when reconstructing outbreaks of the pine looper. The history of outbreaks reconstructed by the technique for the West Siberian Plain’s forest-steppe includes 11 defoliation events from 1914 to 2017. The results obtained using the developed method are useful to better understand the patterns of population dynamics of the pine looper and other phyllophagous pests.

**Abstract:**

The pine looper *Bupalus piniaria* is one of the most widespread phyllophagous insect species across Northern Eurasia, defoliating Scots pine forests over vast territories. Since there are not enough long-term documented observations on a series of outbreaks, there is a need for methods allowing them to be reconstructed to study their dynamics patterns. Previously, dendrochronological methods were successfully used to solve such issues. However, the most common approach is not applicable for the Western Siberian forest-steppe since it requires comparison with a non-damaged tree species close to pine in terms of longevity and resistance to rot. In the pine forests of the steppe and forest-steppe zones of Western Siberia, there are no species that are not damaged by the pine looper that meets these requirements. Methods allowing not using control species are also not free from disadvantages (e.g., weak specificity). Therefore, we have developed a new method based on the analysis, not of the tree-ring width but the early- and latewood width to reconstruct past defoliation events. The past defoliation by the pine looper is indicated by the presence of a negative pointer year for latewood, followed by a negative pointer year for earlywood in a subsequent year among the majority of individuals. Linear modeling showed a difference between the climate impact on radial growth and the defoliation one. The obtained reconstruction was compared with the results of other methods (mowing window, OUTBREAK, independent component analysis), literature, and Forest Service data. The developed new method (pointer year method; PYM) showed high efficiency confirmed by results of the tree-ring series analysis (11 revealed outbreaks in the past). Compared with other reconstruction techniques under the given conditions (a favorable combination of heat and humidity; probably low-intense and short defoliation), the proposed method provided more precise results than those proposed earlier. Due to high accuracy, the PYM can be useful for detecting late-summer and autumn past defoliations of tree species with clear difference between early- and latewood even though the damage was weak.

## 1. Introduction

A considerable amount of both applied and fundamental research is dedicated to the pine looper *Bupalus piniaria* (L.), a mass defoliator of Scots pine *Pinus sylvestris* L. In particular, a study of the temporal dynamics of the *B. piniaria* populations is popular, being of both fundamental (understanding the mechanisms of population dynamics) and applied (prognosis of population dynamics and outbreaks) value [1].

The study of population dynamics should lie upon long-term time series. There are several relatively long time series for *B*. *piniaria* created at different times for the North European Plain [2,3], the East Coast of Scotland [4], and the Minusinsk Hollow [1]. However, such studies are lengthy and laborious, and in some cases, it is possible and effectual to use effects caused by needle losses [5,6]. 

Pine looper damages pine forests in the West Siberian Plain’s (WSP) forest-steppe zone over vast areas (up to tens of thousands of hectares) [1]. The density of population in the forest-steppe and steppe zones of WSP is high for Siberia, therefore the total or partial death of affected stands causes substantial economic damage. Populations of some other economically important phyllophagous insects behave similarly, for example, the Siberian silk moth *Dendrolimus sibiricus* (Tschetv.) in Asia [7], winter moth *Operophtera brumata* (L.) in Europe [8], and Douglas-fir tussock moth *Orgyia pseudotsugata* (McDunnough) in North America [9]. E.N. Palnikova and colleagues’ monograph [1] describes the most detailed history of the pine looper outbreaks in the forest-steppe of the WSP. However, the data summarized there relate mainly to the forests of northern and central parts of the region and cover the period from the 1930s to the beginning of the 2000s. The *B*. *piniaria* outbreaks data relating to the lower reaches of the Biya River (eastern part of the same region) is fragmentary [1,10]. Meanwhile, the local pine looper population is a suitable model object for studying the population dynamics of both this species and other eruptive phyllophagous pests since fluctuations in their abundance have some common patterns [11]. It is known that in the pine forests of the northern and central parts of the forest-steppe zone of the WSP occur outbreaks of nun moth *Lymantria monacha* (L.) and pine beauty *Panolis flammea* (Denis and Schiffermüller). However, the Biysk pine forest (southeastern part of WSP’s forest-steppe) was not infested by these species [10]. The absence of other pine defoliators relieves us of the need to differentiate their impact on radial growth.

Considering the absence of documented data, pest outbreaks history could only be reconstructed to varying degrees of reliability. Probably, the only way to perform such a reconstruction is to use tree-ring chronologies for the affected species. Dendrochronological reconstruction can confirm and supplement the literature source data, and most importantly, extend the history of outbreaks into the past. Due to the small number of objects suitable for research (rather old stands repeatedly damaged by insects), such studies are rarely published. Nevertheless, there is some experience in reconstructing outbreaks using dendrochronological methods, mainly for North America and the Alps (e.g., [12,13,14,15]) but less frequently for central Asia [16]. All of the above and other research dated defoliation events had occurred before the start of regular observations of the phyllophagous insects’ abundance. 

The reference approach to outbreaks reconstruction involves comparing the chronologies of tree species damaged by a studied insect with non-damaged ones. Still, a comparison with other series (e.g., climatic) is also possible [15]. However, in the forest-steppe of the WSP, such an approach can hardly be implemented. The absence of tree species comparable to pine in terms of longevity and resistance to rot [17] does not allow constructing long tree-ring series to be used as an undamaged control. Other time series having an annual resolution (climatic, hydrological) are also not long enough. For this reason, we turned to a rarely used approach based on the analysis of the tree-ring series’ characteristics specific for defoliation but not for other disturbances [6,13]. Such methods described in the literature [6,16,18] are based on the analysis of the tree-ring width. However, this characteristic of radial growth could be negatively influenced by other factors such as droughts [6] or forest fires [19]. 

The differentiation issue can be solved by using the seasonwood increment to reconstruct the defoliation history. We assumed that the seasonal growth response differs depending on the month of defoliation. For instance, damage by the gypsy moth *Lymantria dispar* L. in the middle of summer does not affect the earlywood formation in the year of defoliation but reduces the latewood formation; earlywood becomes narrower in the next season only [20]. It is well known that the radial growth rate does not much change the year *B. piniaria* defoliated a tree because the highest defoliation rate is reached after needle growth and wood formation were both completed and carbohydrate reserves assimilated [1]. Earlywood of the post-defoliation year is formed using mostly assimilates of the previous season [21,22,23], that is why previous year defoliation also does not much affect current year earlywood growth. High first-order autocorrelation indirectly indicates a strong effect of the past (defoliation) year on the current year’s growth [23], which includes the active use of reserves accumulated in the previous season. Latewood is composed mainly of carbohydrates produced during the current year photosynthesis [21]. Considering that defoliation diminishes tree photosynthetic capacity, latewood growth in the next year after defoliation is sharply and synchronously decreased in most of the damaged stand. For the same reason, the reserve of carbohydrates decreases, which leads to a sharp and synchronous decrease in the growth of earlywood in the subsequent year (the second year after defoliation). The fact that insect defoliation at different timing affects early- and latewood formation was previously shown by the example of *Lymantria dispar* (L.) for common oak *Quercus robur* L., *Zeiraphera griseana* (Hübner) (=*diniana* Guenée) for the European larch *Larix decidua* Mill. [24] (deciduous), and *Coloradia pandora* Blake for the ponderosa pine *Pinus ponderosa* Dougl. ex Laws. [6] (evergreen). N.A. Straw previously noted a decrease in the proportion of latewood due to pine looper defoliation in Scots pine [4]. Thus, it is possible to indicate damage caused by mid-summer–early-autumn defoliators using the gap between late- and earlywood growth suppressions.

Before using this approach for cases when it is necessary to separate the effect of defoliation by the pine looper from other factors (other phyllophagous insects, fires, etc.), the methods’ efficiency should be tested and decided whether it gives an advantage before previously developed methods. The present study attempts: (a) to develop a method for reconstructing outbreaks of the pine looper, based on the analysis of the seasonal wood increment; (b) to compare it with previously developed methods for detecting defoliation in tree-ring series, which are not based on comparison with the non-damaged tree species chronologies; (c) to reconstruct the history of outbreaks of the pine looper in the lower reaches of the Biya River. After data processing, we faced an additional issue—the optimal parameter values of the algorithms previously proposed for detecting defoliation signals in tree-ring series differ significantly between the original publications [6,15,17] and our results.

## 2. Species Background

The pine looper is monophagous to Scots pine *Pinus sylvestris* L. throughout its entire range. This species classified as an eruptive one, which means a high level of variation in characteristics (primarily abundance) within a population over time. As these changes occur, a population undergoes a series of sequential qualitative transitions determined by intra-population processes, which define the quasi-periodicity of abundance fluctuations [11]. The periodical rise of *B. piniaria* abundance causes severe damage in pine forests [1,3] throughout Eurasia from North European Plain and Scotland to Transbaikalia, including Biysk pine forest where the pine looper is the only mass Scots pine defoliator [1,10]. 

The pine looper is a univoltine species. The flight of adults and oviposition take place after overwintering as pupae. The flight period starts above a threshold of a daily mean temperature of 15 °C (growing degree days 320 °C, base temperature 4.9 °C). Larval feeding ends quite late in autumn when the temperature approaches 0 °C. Most intensive foraging of *B. piniaria* occurs in late summer–early autumn; that is why the species belongs to the autumn phonological group. The pine looper overwinters as a pupa on the forest floor [1].

## 3. Materials and Methods

### 3.1. Study Area

Samples have been collected in stands of the lower reaches of the Biya River (southeastern part of the WSP). The stands are located in the Biysk pine forest—a relatively small forest area stretching along the Biya River, and in the nearest southeastern part of the Upper Ob pine forest sharing similar characteristics (in the present study, they are combined under the name Biysk pine forest) (Figure 1). The study area is located on the territory of the Biysky and Zonalny regions of the Altai Krai (southeastern part of WSP) that are classified as forest-steppe. According to the relief features, the study area can be classified as flat land. The climate of this territory is continental. By the nearest weather station Biysk-Zonal’naya (52.69° N, 84.93° E), the mean annual temperature is 1.8 °C. The warmest month is July, with a mean temperature of 19.6 °C. Period of warmth with temperatures rising above 0 °C last for 207 days [25]. According to the average long-term data, the temperatures of late May–early June is enough for the beginning of *B*. *piniaria* mass flight and oviposition [1]. The autumn transition below 0 °C coincides with the end of October [25]. Such conditions are optimal for *B. piniaria* development [1].

### 3.2. Selection of Stands and Trees

Collection of tree-ring cores was conducted in stands (Table 1) with the absolute predominance of pine. The recreational load is high in the Zarech’e site and most significantly in the Lesnoye site located on the outskirts of a village, but heavy anthropogenic damage, such as cutting, bark stripping, etc., is absent. The recreational load in the Sokolovo site is minimal. No evidence of severe fires (e.g., fire scars on tree trunks) was found in the study plots. According to Forest Service data, the studied forests were damaged by one or two outbreaks of the pine looper in 1996–1999 and (or) in 2003–2008. Trees chosen for sampling are dominant and have no signs of mechanical damage or rot.

### 3.3. Tree-Ring Preparation, Measurement, Crossdating, and Standardization

Dried cores were glued in wooden support and sanded manually by progressively finer abrasive paper up to 400 grit. Before measuring, they were checked for avoiding of compression wood and other irregularities. Measurements of tree-rings, early- and latewood were performed on scanned images (1200 dpi or more) using the CooRecorder program (Cybis, Saltsjöbaden, Sweden) [26].

Rejection of samples that are not typical for the studied area and identification of missing or false rings were performed by cross dating the samples with the CDendro software (Cybis, Saltsjöbaden, Sweden) [26]. At first, the tree-ring series were standardized using the Baillie and Pilcher procedure [27], then checked by the leave-one-out method. If the Pearson *r* between tests series and generalized (without tested series) site chronology turned out to be <0.4, the series were not used in the research [26]. In case several tree-ring series built for one tree were included in the further processing, these series were averaged to obtain a single series for each tree. After the rejection of all series, which radial growth differs from the general pattern, the chronology confidence was evaluated with the expressed population signal (EPS) and the signal-to-noise ratio (SNR). The reliable time span was defined by a threshold of EPS ≥ 0.85 [28] using the moving window (20-year period, 10-year overlap) using the dplR package [29] in the R environment [30].

### 3.4. Reconstruction of Outbreaks History

#### 3.4.1. Pointer Year Method

It was previously shown that the width of earlywood and latewood respond differently to defoliation [6,20]. It can therefore be expected that if the damage caused by these species of phyllophagous insects is confined to a particular season, then: (a) the growth of early- and latewood will change in different ways, and (b) these differences will be the same for any of this species outbreaks. The reconstruction technique proposed here is limited to the late-summer and autumn phenological group of pest insects.

It follows from the scheme (Figure 2) that past defoliation by *B*. *piniaria* can be detected when searching for pointer years [31] in tree-ring series built for early- and latewood simultaneously. Here we propose a method of past outbreaks detection based on this approach. As a criterion for identifying pointer years, we used the relative growth change (*RGC*) [32], calculated (for early-, latewood, or ring width) as the ratio between current year absolute radial growth *R_t_* and growth of the previous one *R_t−_*_1_:(1)$$RGC_{t\left(EW,LW,width\right)}=\frac{R_{t\left(EW,LW,width\right)}}{R_{t-1\left(EW,LW,width\right)}}100\%\;.$$

The year *t* was considered a defoliation year if: (a) the year *t* + 1 latewood *RGC* value was less than the threshold *change.l* (%) for the proportion of trees equal to or greater than *sync.l* (%) and (b) the year *t* + 2 earlywood *RGC* value was less than the threshold *change.e* (%) for a proportion of trees equal to or greater than *sync.e* (%) (Figure 2). The method for recognizing past outbreaks based on this approach will be referred to as the pointer year method (PYM).

#### 3.4.2. OUTBREAK Method

A more common approach to reconstructing outbreaks history is based on the annual ring width patterns of affected trees. It is a well-known fact that a more or less prolonged recovery follows the drastic decline in radial growth caused by defoliation. The history of insect outbreaks reconstructions based on the detection of such patterns in the series of annual ring width measurements is implemented in the OUTBREAK algorithm [9] (hereinafter, this approach is referred to as the OUTBREAK). A version of this algorithm for cases when it is impossible to use an unaffected tree species’ radial growth as control is described by James H. Speer and colleagues [6].

The algorithm’s initial data included the detrended by cubic smoothed spline ring width series for each tree in the studied area [9,28,29]. The assumed defoliation was recorded by the algorithm for an individual tree in year *t* if: (a) *t* and following years in the number ≥ *lng* have got radial growth indexes ≤ (*std* × standard deviation of series) and (b) *RGC* (Equation (1)) for the year *t* ≤ *abrupt*. We would consider the year *t* as the first year of defoliation if the proportion of trees meeting both of these criteria was ≥10%.

#### 3.4.3. Mowing Window Method

Several studies considered the ring width’s local minimum values as an additional sign of defoliation [16,33,34]. This method implies comparing absolute ring width in *t* year with ones on the interval from *t* − *width* to *t* + *width* years. If the proportion of trees with the local minimum in *t* year is ≥*perc* (a unit fraction), this serves as evidence in favor of probable defoliation in the year *t* [16,34]. From now on, this approach is referred to as the moving window method (MWM).

#### 3.4.4. Method of Independent Component Analysis

The method of outbreak reconstruction based on multivariate statistics is known [18]. The tree-ring series studied using the method are preliminarily reduced to the same length by removing their early sections, which is necessary for the method to work correctly. Next, the series of measured annual ring width values are standardized so that the arithmetic mean of each of them becomes equal to zero, and the standard deviation becomes equal to one.

The resulting matrix of the annual ring width indexed values is decomposed by the method of independent components fast analysis into separate components using the ica package [35] in the R environment. Evidence of defoliation (or other unfavorable effects) is recorded preliminary if values of any identified component decrease below a certain value *lim* (threshold value of deviation from the mean) for *lng* (length of such series) years in a row.

On the next step, the ring-width series for each tree were detrended using cubic splines [28,29], and the resulting radial growth indexes record was averaged for the site. If year *t* has been selected on the previous step and its radial growth index declined below some quantile *q*, this coincidence was interpreted as a consequence of defoliation [18]. From now on, the method is called MICA (method of independent component analysis).

Besides, some artifacts might appear during the preceding calculations. It is possible that by these artifacts, the minima in detrended chronology will not coincide with low values in the independent components. However, we assumed that such an offset should not exceed the *rng* value.

#### 3.4.5. Optimization of Algorithms Parameters

In the protocols we used [6,9,18,34], there are no justifications for specific values of arguments. Nevertheless, it is not obvious that values that had been used previously are optimal for our case. Therefore, we try to found optimal values of the arguments using PYM, MWM, MICA, and OUTBREAK methods for each investigated sample plot. For the first three methods, values were considered optimal if they had indicated outbreaks of 1996–1999 and 2003–2008, known from the Altai Forest Protection Center’s departmental materials, with a minimum (ideally in the absence) of false-positive results. The OUTBREAK method’s optimality was assessed by maximizing the proportion of trees indicated as defoliated during the *B. piniaria* outbreaks mentioned above.

### 3.5. Correspondence of Reconstruction Results

Identifying past defoliation events using OUTBREAK, MWM, and MICA requires searching for the radial growth’s local minimum values, which appear several years after defoliation (for more information, see Section 5). Therefore, the PYM results were considered confirmed by the other methods data if they detected the radial growth reduction in the year of defoliation assumed by PYM or from one to four years after it. To assess the contingency between the results obtained using the PYM and other methods, we analyzed the uncertainty matrices using Fisher’s exact test *F*, correcting significance levels using the Benjamini and Yekutieli method [36,37] (Table 2):

Additionally, using these matrices and the caret package [38] for R, the following correlation characteristics were assessed: accuracy (*acc*; the probability of coincidence between the results of PYM and other methods), sensitivity (*sens*; resistance to a negative result when the PYM result is positive), and specificity (*spec*; resistance to a positive result in the absence of defoliation according to the PYM data).

### 3.6. Climate and Outbreaks

We use linear modeling for evidence of the *B*. *piniaria* defoliation’s substantial impact on radial growth in studied stands. The tree-ring series for this purpose were subjected to double-detrending (negative exponential curve then cubic-smoothing spline detrending) and averaged for each site separately [28].

First, we select climate characteristics (average month temperature and total month precipitation; obtained from Biysk-Zonal’naya weather station) for modeling using a response function [28,39]. The response analysis was carried out in the bootRes package [40] for the period from January of the previous year to September of the year when the tree ring was formed. Selected characteristics had to demonstrate a statistically significant response at 0.05 level.

A preliminary study of the ring width data by both observations and superposed epoch analysis (SEA) [29] revealed that the radial growth decline following defoliation by the pine looper is short (one–two, sporadically three or four years) in the studied areas. Thus, we were forced to reject the previously proposed approaches to separate the effect of weather and defoliation on the annual ring formation [41,42] since they are most suitable for studying long-term (four years or more) growth reduction.

To evaluate the effects of the pine looper defoliation on variation in radial growth, we constructed linear models of the relation between tree-ring width and climate conditions for each site, disregard (fixed effect model; Equation (2)) and considering (dummy variable model; Equation (3)) the effects of the outbreaks:(2)$$I_{t}=\overset{n}{\underset{i=1}{\sum }}a_{i}W_{it}+C,$$
(3)$$I_{t}=\overset{n}{\underset{i=1}{\sum }}a_{i}W_{it}+bO_{t}+C,$$
where *I_t_*—radial growth index in year *t*; *a_i_*—coefficient for the *i*-th climate characteristic; *W_it_*—value of the *i*-th climate characteristic in year *t*; *b*—coefficient of the dummy variable *O_t_*, which indicates the absence (*O_t_* = 0) or presence (*O_t_* = 1) of defoliation effects in the year *t*; *C*—intercept. We considered a sharp decline in tree-ring width occurring one–three years after damage as a symptom of the significant effect of defoliation in this year. The final choice of predictors was based on the backward stepwise selection by the Akaike information criterion.

The correspondence of simulation results to real data was assessed using the adjusted coefficient of determination $R_{adj}^{2}$. The statistical significance of the differences between $R_{adj}^{2}$ for models including and excluding defoliation was tested by bootstrapping. For each model, 100 bootstrap samples were created containing data on the previously selected climate and defoliation characteristics. Further, for these samples, models were built according to Equations (2) and (3). At the final stage, we compared the $R_{adj}^{2}$ values obtained for the models including and not including defoliation as a predictor. Since the modeling was carried out on linked bootstrap samples, Friedman’s test was used for statistical comparison.

## 4. Results

### 4.1. Tree-Ring Series Characteristics and Optimal Values of Outbreak Detection Parameters

After crossdating, only seven cores (three trees) collected in Sokolovo were excluded from further processing. Eight missing rings were found (no more than one ring per tree) in Sokolovo. Based on the moving window EPS analysis results, ca. 70–100-year periods for further processing were selected (Table 3). The high EPS and SNR values calculated for these plots confirm the representativeness of the taken samples. 

The optimal parameter values (in terms of precise prediction of well-documented 1996–1999 and 2003–2008 outbreaks) were found for all stands and defoliation history reconstruction algorithms used in the research (Table 4).

### 4.2. Pine Looper Outbreaks Reconstruction

#### 4.2.1. Zarech’e

The PYM’s use with the optimal argument values (Table 4) showed that the defoliation by the pine looper in the Zarech’e site occurred in 1949, 1959, 1972, 1978, 1986, 1996, and 2001 (Figure 3). The OUTBREAK algorithm confirmed all these defoliation events, while the MWM confirmed only five ones. Since the MICA requires the identical length of the studied radial growth series, data up to 1967 were excluded from the analysis (23 years, 33.3% of the time series length). In the interval from 1967 to 2013, four from five PYM-detected outbreaks coincided with MICA-detected ones (Figure 3).

#### 4.2.2. Sokolovo

At the Sokolovo site, for which the most extended period was analyzed (Table 3), the PYM discovered the consequences of defoliation by the pine looper in 1915, 1924, 1930, 1941, 1949, 1972, 1984, 1995, and 2002 (Figure 4). Unlike Zarech’e, only eight from nine outbreaks identified by PYM, were detected by OUTBREAK. However, all these outbreaks were followed by local growth minima detected by MWM. The length of the time interval analyzed by MICA coincided with the other methods’ results, but it confirmed only five out of nine defoliation events detected by the PYM (Figure 4).

#### 4.2.3. Lesnoye

According to the PYM results, the pine looper defoliated the Lesnoye site stand five times (1949, 1959, 1978, 1990, and 1995) in the studied time interval (Figure 5). Each of the defoliation events detected by the PYM corresponds to a radial growth decline identified by the OUTBREAK. The MWM confirmed the defoliation events identified by the PYM only in three cases out of five. For the correct MICA application, the studied time interval was reduced by three years (3.7% compared to its duration for the other methods); all assumed defoliation events were included in this interval. Only two out of five outbreaks reconstructed by the PYM were confirmed by the MICA results (Figure 5).

### 4.3. Compliance of Reconstructions Obtained by PYM and Other Methods

#### 4.3.1. OUTBREAK

The OUTBREAK showed the most stable results in terms of the correspondence to reconstruction based on the PYM (Table 5). Along with high-significant Fisher’s contingency, we see very high *acc*, *sens*, and *spec* values. This fact shows both a close correlation between PYM and OUTBREAK results and good resistance of OUTBREAK to false-positive (negative) results. Most of the sizable and prolonged growth declines, which have been detected by OUTBREAK, begin simultaneously with the alleged defoliation or a few (up to two) years later (Figure 3, Figure 4 and Figure 5).

#### 4.3.2. Mowing Window Method

The results of PYM and MWM contingency tables analysis (Table 5) were unexpected to some degree. Surprisingly, even *sens* values were not very high because of some false-negative results in Zarech’e and Lesnoye (see above). Consequently, *acc* values for any given site were lower than other methods’, and the significance of Fisher’s test was worse in comparison with OUTBREAK. Moreover, false-positive results are usual for MWM (low *spec* values). Lags between local minima detected by MWM and PYM-detected defoliation events are usually two, rarely three or four years (Figure 3, Figure 4 and Figure 5). 

#### 4.3.3. Method of Independent Component Analysis

The MICA results are generally consistent with the same for PYM (Table 5). This method is characterized by the highest resistance to false-positive results (see *spec* values) and acceptable accuracy. However, MICA has shown such a significant drawback as a deficient level of true-positive results (low *sens*). It is confirmed by the results of Fisher’s exact test for the Lesnoye site. We have experienced difficulties estimating the lag between the start of PYM- and MICA-detected events. It is connected to both overlapping between periods of decrease independent component values and a long series of consecutive low growth indices (Figure 4). Nevertheless, the fact of the presence of lag between PYM and MICA results is indisputable. 

### 4.4. Effects of Weather and Defoliation on Tree Ring Formation

The period of suppressed growth after an outbreak is two years for all sites, as shown using SEA (significance level *p* < 0.05). 

When modeling radial growth using only weather characteristics, the $R_{adj}^{2}$ value varied from 0.211 (Sokolovo) to 0.444 (Lesnoye) (Table 6). In general, there is a statistically significant at 0.05 level relationship (in terms of response analysis) between the ring width and precipitation of the beginning (May–June) and temperatures of the end (June–August) of the growing season of the current year. There is also a correlation with the previous year’s winter conditions (January temperatures or precipitation) (Table 6).

The effect of defoliation on tree-ring width formation is relatively small: $R_{adj}^{2}$ increased by 0.058–0.194 when this factor was taken into account (Table 6). However, the bootstrapping confirmed the statistical significance of the observed changes. The significance level in all cases was below 0.001.

## 5. Discussion

### 5.1. Pine Looper Outbreak Reconstruction

#### 5.1.1. Comparison of Climate and Defoliation Contribution to Radial Growth

The reconstruction of defoliators’ outbreaks history is possible only if the effect of defoliation is not hiding by climate impact. The prolonged defoliation impact, as shown by SEA, is short-term in this region. Statistically significant growth suppression duration is two years, in some cases up to four years (e.g., 1941 outbreak followed by 1945 local minima; see Figure 4). The impact of needles loss (in terms of improvement of the model) is comparatively small. The advantage of including the defoliation factor in our model is commensurate with [42] and far less than in [41,43]. Nevertheless, the statistical significance of the differences between climate only and climate/defoliation models is very high for all three sites. Thus, the modeling shows that outbreak consequences had made an apparent contribution to radial growth of Scots pine in the Biysk pine forest.

#### 5.1.2. Consistency of Pointer Year Method Results with Historical Data

Eleven pine looper outbreaks were identified in the territory of the Biysk pine forest using PYM, while six outbreaks were previously described [1,10,44] and documented by the Forest Service (Figure 6). The results of PYM defoliation dating do not match with data previously recorded and documented by the Forest Service with an accuracy of up to a year (Figure 6 and Figure 7). Nevertheless, this does not mean the reconstruction is erroneous. Firstly, depending on the particular stand characteristics, the highest phyllophagous insects abundance is reached at different times, which is also shown for *B. piniaria* [45]. Initially, the insects defoliate sites with preferred characteristics (in the case of pine looper, these are internal parts of high-density 20–40-year-old pine forests [46]) and then expand into new territories [11]. Therefore, damaged stands are spatiotemporally inhomogeneous, and the minimum ring width in some stands is reached earlier than in others. Secondly, the defoliation impact on radial growth, at least in some cases, became apparent earlier than the maximum needle loss is reached (see [47] Table 1, [48] Figure 4), which is most commonly identified as a defoliation period. Finally, the sources studied have no criteria for indicating both the outbreak’s beginning and the points where it was recorded. Under given circumstances, it cannot be assumed that the reconstruction results will exactly match with the periods during which, according to the literature and Forest Service data, the abundance of *B. piniaria* reached its maximum.

For the Zarech’e site, located within the investigated pine forest (Figure 1) and having no signs of heavy anthropogenic or natural disturbances (cutting, severe fires, intensive recreation, etc.) after current tree generation’s onset, the defoliation peaks reconstructed using the PYM are one–two years ahead or coincide with the outbreaks beginning indicated in the sources (Figure 7). The Sokolovo site is also undisturbed but located on the western edge of the Biysk pine forest (Figure 1). In our opinion, this is the reason for the delay of defoliation events reconstructed using the PYM, which was recorded for two outbreaks of *B*. *piniaria* (Figure 5) since the maximum abundance stage was first reached in the inner parts of the stand, and then defoliation affected its periphery. The Lesnoye site is located both on the geographical and ecological (due to the proximity of the settlement) periphery of the Biysk pine forest (Figure 1). Because of edge effect and recreation, the impact of exogenous processes, such as cyclic forest insects (including the pine looper) population fluctuations, is weakened. Therefore, some of the outbreaks (1940–1942, 2003–2008) did not affect this site. Moreover, the defoliation during the 1987–1990 outbreak coincided with the end of the period recorded (Figure 5). It can thus be stated that both the mismatch between the reconstructed dates of defoliation beginning and the presence of undamaged forest fragments during a particular outbreak is typical of phyllophagous insects. For example, despite the relatively small distances between sites, this has been described for *C*. *pandora* [6] and *Choristoneura occidentalis* Freeman [49].

It is important to point out the coincidence of *B*. *piniaria* outbreaks in Biysk pine forest and forested area dominated by *P*. *sylvestris* situated about 250–400 km westward (Figure 6). We cannot statistically analyze this phenomenon due to gaps in data, but available information allows us to assume high coherency between pine looper populations in these regions, possibly because of the Moran effect [50,51,52].

### 5.2. Consistency between Pointer Year Method and Other Algorithms of Outbreak Detection

#### 5.2.1. Potential of Analyzed Algorithms in the Local Conditions

The OUTBREAK method has shown better results (compared to MICA and MWM) when reconstructing the *B. piniaria* outbreaks history in the Biysk pine forest. It is surprising because the version of this algorithm we used was developed and tested for *C. pandora*. This species featured drastic and prolonged growth reduction after their outbreak [6]. On the contrary, pine looper outbreaks in the forest-steppe’s mild climate (see Section 5.3.1) lead to a comparatively weak and short growth decrease. Nevertheless, the OUTBREAK algorithm showed better correspondence with the more sensitive pointer year method.

Another method tested for *C. pandora* outbreaks detection is MICA [18]. It showed encouraging results for the *C. pandora–P. ponderosa* system, but in the local variant of defoliator–host-tree pair, this method is very unstable (Table 5). We have no assumptions about the causes of this failure. Possibly, it related to some details of independent component analysis [35] itself.

We were not expecting the false-negative results of MWM (Table 5, Figure 3 and Figure 5). It is an auxiliary but robust method of past outbreak detection [16,33,34]. The most probable reason for this error is the uneven rate of trees reaction to needles loss. For example, after the defoliation of the 1959 year in Zarech’e, the proportion of trees with a local minimum of ring width was 0.261 in 1962 and 0.333 in 1963. However, these proportions are below the optimal threshold (0.35; see Table 4). Thus, nominally, this outbreak is not followed by a local minimum of the radial growth, although the sum of these proportions (0.594) is considerably higher than the accepted threshold. For the prevention of similar artifacts, the MWM method required some improvement.

#### 5.2.2. Source of Lags between Pointer Year Method and Other Algorithms

When reconstructing the forest insect outbreaks history, an obvious criterion determining the quality of reconstruction is the accuracy of identifying the defoliation period. In this regard, attention is drawn to the not quite exact coincidence of the results of the methods used in the research (Figure 3, Figure 4 and Figure 5).

We associate the lags between the results of the PYM and other methods with the fact that they are used to study different objects, namely, a specific pattern of early- and latewood declines in the case of PYM and the decline in a tree-ring width for other methods. It follows from the conceptual scheme underlying the PYM (Figure 2) that the signal detected by this method appears in the radial growth series strictly during two seasons following defoliation, which allows dating it with a minimum error. The accuracy of defoliation dating using other methods is significantly affected by the inertness of the tree-ring width change after the loss of needles. It has been repeatedly shown [20,24,53,54,55,56], for the *B. piniaria* as well [1,4], that the most significant tree-ring width decrease occurs one–two years after defoliation. It was confirmed for our data by SEA too. Furthermore, the duration of the delay may depend on the degree and frequency of defoliation [48], the location of trees at the edge or within the stand, their age [20], in other words, variables, which values, or their effect on radial growth, are difficult to evaluate retrospectively. It thoroughly explains why the reconstructions built using OUTBREAK, MWM, and MICA, which are based on the search for local radial growth minima, are somewhat lagging behind the PYM reconstructions.

### 5.3. Possible Reasons for the Mismatch between the Values of the Arguments for OUTBREAK, MWM, and MICA in the Present Research and Previous Studies

When choosing the optimal values of the arguments for the methods used in the *B. piniaria* outbreaks reconstruction (Table 4), we faced the fact that they differ significantly from those indicated by our predecessors. For example, to search for evidence of defoliation of the ponderosa pine *P*. *ponderosa* by the pandora moth *C*. *pandora* using the OUTBREAK algorithm [6], the following values were used (hereinafter, the identifiers of arguments accepted in this study is used): *std* ≥ 1.28, *lng* ≥ 4, *abrupt* ≥ 0.5. In other words, after defoliation of Scots pine by the pine looper in the Biysk pine forest, the decline in growth was less sharp, deep, and prolonged compared to the outbreak of *C. pandora* in the Cascade Mountains on *P. ponderosa*. There are also significant differences for the argument *lim* of the MICA algorithm: In the article describing it [18], the past defoliation of *P. ponderosa* by the Pandora moth is stated at *lim* ≤ −1.6, which also indicates a more profound decline in growth compared to the Biysk pine forest (Table 4). For the MWM [16], the mowing window is much wider (*width* = 7). This method was used to reconstruct the *Larix* species’ defoliation history by larch tortrix *Z. griseana* in the Alps and unspecified pests (hypothetically, *Z. griseana* and *Z. lariciana* Kawabe or the lappet moth *Cosmotriche saxosimilis* De Lajonquière) on the Tibetan plateau [16,34]. The *perc* values adopted as the optimal ones for MWM (characterize the synchronicity of the radial growth decline) are higher than in the cited publications [16,34]. In general, the authors of the previous studies [6,16] presume more long-lasting effects of defoliation (arguments *lng* for the OUTBREAK and *width* for the MWM). Although it is difficult to compare data on the duration of the radial growth decline after defoliation from different sources due to the differences in its assessment methods, some patterns can still be found.

#### 5.3.1. Climate

The first possible reason for discrepancies between argument values in our research and previous ones is the climatic differences. For Biysk pine forest, we can suppose the absence of additional stresses associated with unfavorable climatic conditions, or, at least, their small impact on defoliated or recovering stands. According to the detailed Köppen-Geiger climate classification [57], a longer (>3 years) radial growth recovery following outbreaks (corresponds bigger OUTBREAK *lng* and MWM *width* values) is typical for arid (B) climate regions [1,58], or at least regions having dry summer (D_s_) [59,60]. Long-term radial growth declines may sometimes be observed in regions with insufficient heat supply [24,55]. It is a consequence of simultaneous and successive stress factors effect, leading to a reduction in total stored carbohydrates, reducing the trees’ ability to regenerate [61]. Biysk pine forest belongs to the region with a milder D_fb_ climate (cold, no dry season, warm summer)—one of the boreal zones gentlest climate types [57]. Recovery after single defoliation may take two-three years for trees in regions that are sufficiently provided with both heat and moisture [4,53], although the recovery period may take much longer [54]. The relatively high humidity and heat supply in the milder climate of Biysk environs [25] reduce the risk of additional stress associated with unfavorable weather conditions, facilitating the return of metabolism to normal [61]. It is especially evident when comparing with pine defoliation by the same species in the Minusinsk pine forest, growing in much more arid conditions [25], that recovered within about ten years [1]. Similarly, D.E. Ryerson and colleagues partly explain the insufficiently distinct effect of defoliation by *C. occidentalis* on the radial growth of *P. menziesii* by more favorable climatic conditions of southwestern Colorado compared to other parts of this insect outbreak area in the southern Rocky Mountains [12].

#### 5.3.2. Degree of Damage

It should be taken into account that the reduction in growth following defoliation depends on its multiplicity [56] and intensity [24]. The apparent assumption that the magnitude of the radial growth decrease depends on the degree of defoliation is confirmed by direct observations [62]. In the Biysk pine forest, the intensity of crown damage by the pine looper can be relatively high, up to the death of tree stands [1]. It is also confirmed by the Forest Service survey’s results, according to which in some cases the population density (the number of wintering pupae per m^2^) achieved or exceeded the value of 6 pieces per m^2^, indicating the threat of complete defoliation [46]. Accordingly, a much deeper radial growth decline was to be expected (*std* arguments for the OUTBREAK, *lim* for the MICA) than it was noted in the present research (Table 4). However, our analysis demonstrates a relatively weak impact of defoliation on radial growth (for comparison, see results for *Z*. *diniana* in [41]). In fact, the duration of the radial growth decline after the documented and suspected defoliation events in the Biysk pine forest indicates a relatively low level of damage during the outbreak of *B. piniaria*, which accelerates the recovery growth rate at the pre-outbreak level [24]. The uneven distribution of insects over the area may influence the obtained results. Indeed, during the abundance increase in 1996–1997, pupal densities of *B. piniaria* ranged from 0.36 to 7.8 pupae per m^2^, and from 0.08 to 33.9 pupae per m^2^ in 2003–2004 (Forest Service data). For a significant part of the affected territories, the population density was much lower than the critical level of 6 pupae per m^2^ [1,3,4], even within plots of several hectares [3]. Hence, the probability of collecting material for dendrochronological studies in relatively weakly damaged areas is very high.

#### 5.3.3. Outbreak Control Measures

Insecticide treatments carried out, at least, during the population density increased in 1950–1955, 1996–1999, and 2003–2008 also decreased both the pine looper population and defoliation caused by it ([1], Forest Service materials). Meanwhile, the loss of ~30% of the photosynthetic apparatus leads to a noticeable decrease in radial growth [63]. In other words, control measures may lead to incomplete population elimination, defoliation degree declining from heavy to low, and diminishing of growth reduction magnitude and duration [9].

#### 5.3.4. Insects and Trees Biology

A more synchronous reach of the local minimum values of the ring width (bigger *perc* for MWM) after pine defoliation by *B. piniaria* in the studied stands than the values given in other sources [16,34] can be explained by the biological characteristics of insects and their host plants. In the research cited above, the consequences of the genus *Zeiraphera* species outbreaks in larch stands were investigated. It is known, however, that for this genus: (a) defoliation of one stand for more than one year in a row is not typical, and (b) after the larval feeding is finished, the trees partially restore the needles in the same season [64]. Apparently, for these reasons, the defoliation by *Zeiraphera* ssp. does not affect the tree-ring width of the majority of larch trees. Damage to Scots pine by *B*. *piniaria* has a more vigorous and more prolonged (including the initial year of the outbreak and subsequent years) impact due to multiple defoliation and slower foliage recovery [1], which causes a reaction of a more significant proportion of trees in the defoliated stand.

## 6. Conclusions

The reconstruction technique for detecting past outbreaks of the autumn phenological group of phyllophagous insects is proposed. The technique is based on the analysis of the early- and latewood growth of damaged trees. This method does not imply the comparison of radial growth of host and non-host tree species. The tree-ring series analysis using the proposed technique indicated 11 outbreaks that occurred over approximately 100 years. The obtained results confirmed or supplemented the available historical data.

The results obtained for the east of the forest-steppe zone of the West Siberian Plain are well confirmed by a set of reconstructions obtained using other methods. At the same time, our proposed method turned out to be more sensitive and/or specific than any of them, at least in this case. However, it can be supported by results of OUTBREAK and moving window methods.

Comparing the default argument values for those methods and the optimal values for the study region revealed a significant difference between them. These differences indicate the less prolonged and less intense effect of defoliation by the pine looper on radial growth under the given conditions. This might be related to the low-intensity damage and the combination of heat and humidity favorable for the restoration of needles.

## Figures and Tables

**Figure 1 insects-12-00090-f001:**
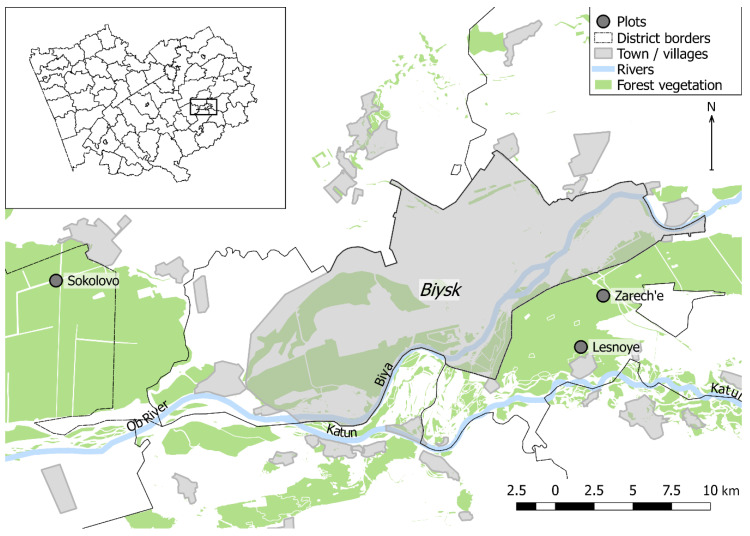
Location of studied forest stands in Altai Krai (insert) and the lower reaches of the Biya River.

**Figure 2 insects-12-00090-f002:**
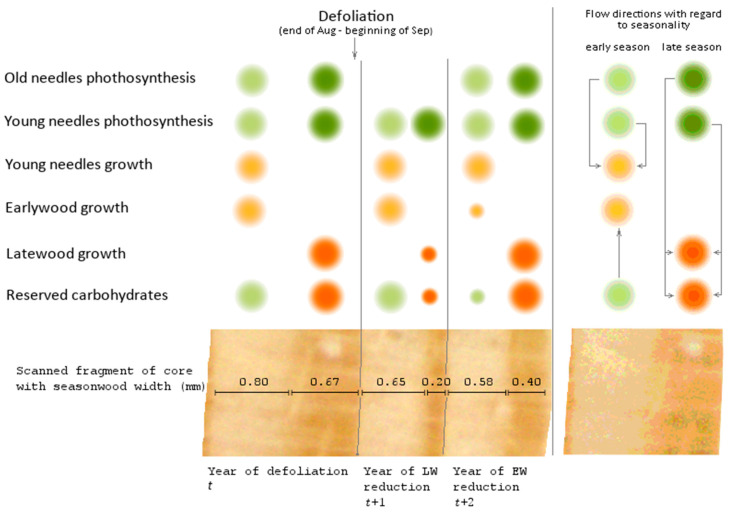
Conceptual scheme of late-summer and fall defoliation impact on assimilates distribution and seasonal wood increment [1,4,6,21,22,23,24]. The sources are green-colored, and the sinks are orange-colored; early season sinks/sources are lighter than the late season’s. The diameter of the circle indicates the volume of synthesized, stored, or consumed assimilates. The principal scheme of carbohydrates flows is located in the right extra section of the figure; the arrows indicate the direction of assimilates relocation.

**Figure 3 insects-12-00090-f003:**
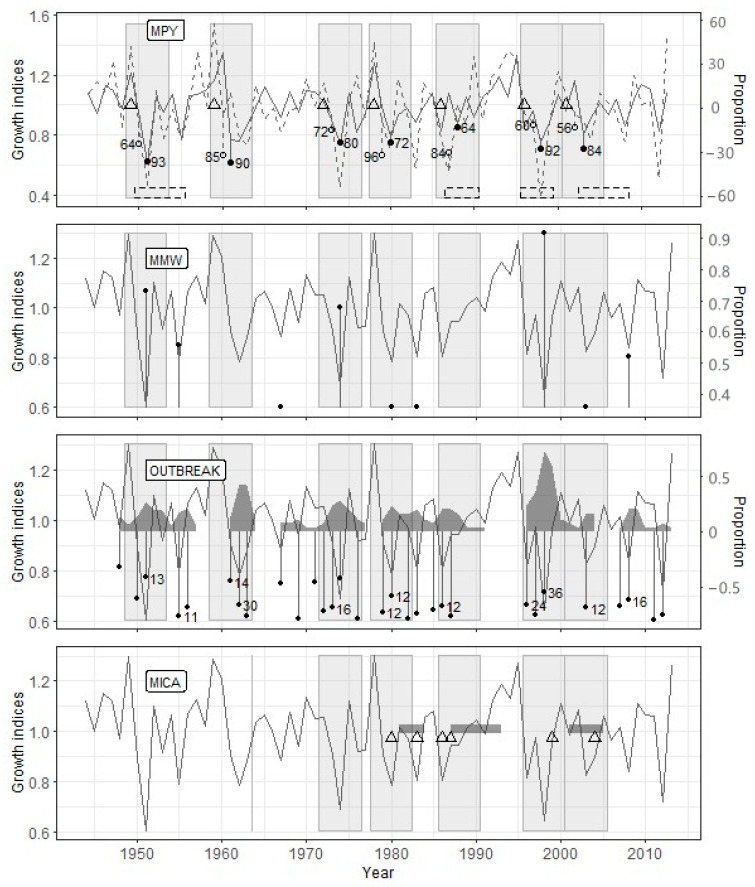
Reconstruction results for the Zarech’e site. PYM: earlywood radial growth index—solid line, latewood radial growth index—dashed line, defoliation year—triangle, *change.l*—open circles (number—*sync.l*), *change.e*—filled circles (number—*sync.e*), periods of documented outbreaks—dashed rectangles. MWM (moving window method): *perc*—filled circles. OUTBREAK: radial growth index decline—filled circles (number—the proportion of trees (%) with radial growth index decline below the threshold), the proportion of trees with a decline in growth ≥ *perc*—grey polygon. MICA: year of assumed defoliation—triangle, the period of a decrease in the values of the selected components below the threshold—dark grey rectangle; to the vertical line’s right is the analyzed time interval. The solid line for MWM, OUTBREAK, and MICA (method of independent component analysis) indicates the radial growth index curve. Light grey rectangles for all graphs indicate the periods of the assumed effect of defoliation on radial growth.

**Figure 4 insects-12-00090-f004:**
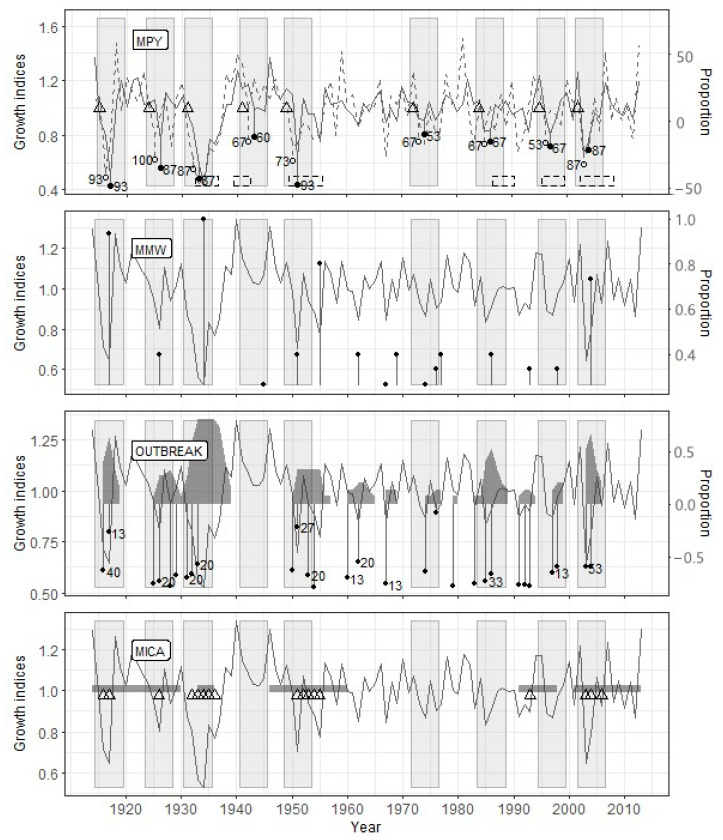
Reconstruction results for the Sokolovo site. Legend as for Figure 3.

**Figure 5 insects-12-00090-f005:**
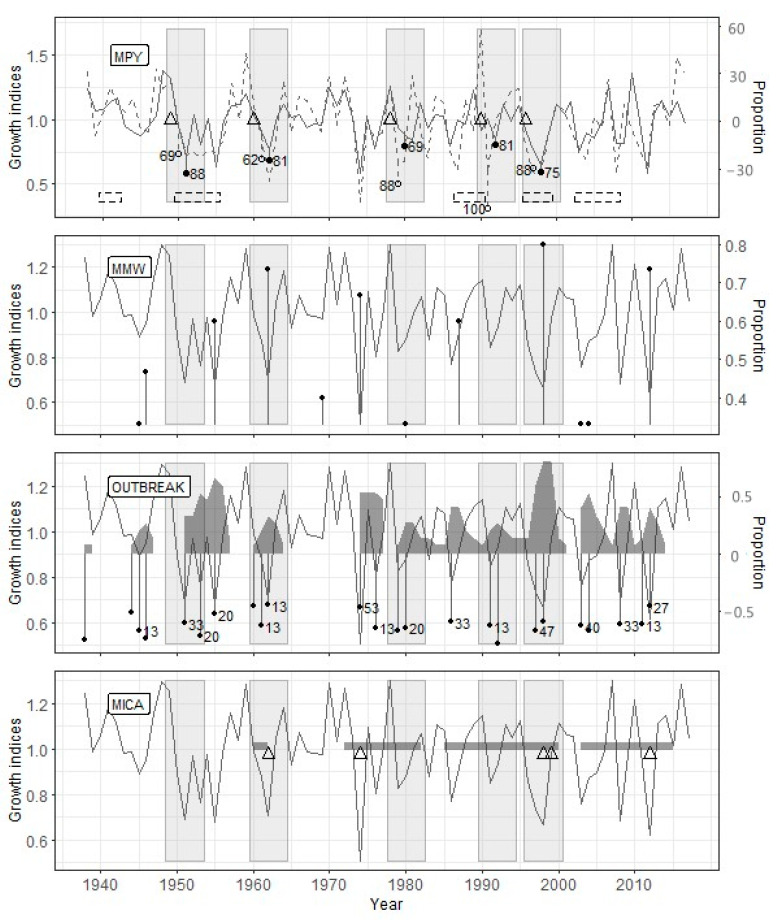
Reconstruction results for the Lesnoye site. Legend as for Figure 3.

**Figure 6 insects-12-00090-f006:**
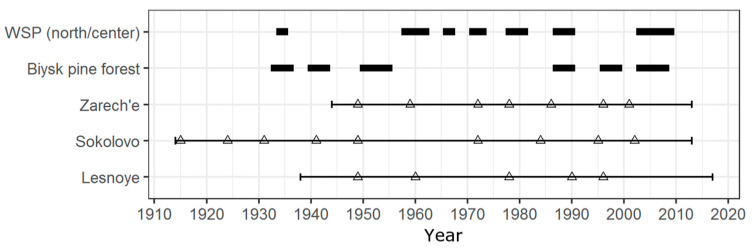
Documented pine looper outbreaks for the northern and central parts of the WSP (West Siberian Plain) forest-steppe [1,10,44], and Biysk pine forest [1,10] (black rectangles), and reconstructed (open triangles) ones. Horizontal lines show the period used for reconstruction.

**Figure 7 insects-12-00090-f007:**
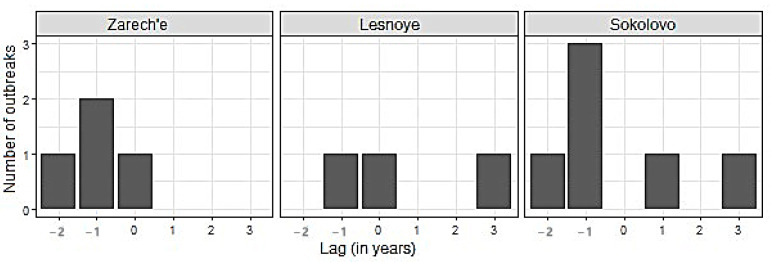
The lag between recorded and reconstructed defoliation periods.

**Table 1 insects-12-00090-t001:** Characteristics of the study sites. Relative density is ratio of real basal area of tree stems to reference one (a unit fraction).

Site	Coordinates	Relative Density	Number of Trees	Number of Cores
Zarech’e	52.51° N, 85.27° E	0.8	25	38
Sokolovo	52.51° N, 84.75° E	0.9	18	36
Lesnoye	52.48° N, 85.25° E	0.6	15	16

**Table 2 insects-12-00090-t002:** The generalized example of contingency table.

		OUTBREAK, MWM, MICA	
		Positive Result	Negative Result
PYM	Positive result	A	B
	Negative result	C	D

**Table 3 insects-12-00090-t003:** Characteristics of the averaged tree-ring chronologies.

Site	Earliest Year	Latest Year	Mean Age	Mean First-Order Autocorrelation	EPS	SNR
Zarech’e	1944	2013	72	0.808	0.975	38.219
Sokolovo	1914	2013	111	0.816	0.960	24.172
Lesnoye	1938	2017	118	0.778	0.968	30.494

**Table 4 insects-12-00090-t004:** Optimal values of function arguments used for *B. piniaria* outbreaks reconstruction.

Site	PYM				MWM		OUTBREAK			MICA			
	*change.l*	*sync.l*	*change.e*	*sync.e*	*width*	*perc*	*std*	*lng*	*abrupt*	*lim*	*lng*	*q*	*rng*
Zarech’e	10	55	10	55	3	0.35	0.6	2	0.8	−1.3	3	0.2	2
Sokolovo	10	50	10	50	4	0.25	0.5	2	0.8	−0.8	2	0.2	2
Lesnoye	15	55	10	65	3	0.30	0.5	2	0.8	−0.9	2	0.1	4

**Table 5 insects-12-00090-t005:** Optimal values of function arguments used for *B. piniaria* outbreaks reconstruction.

Site	Contingency Parameters	OUTBREAK	MWM	MICA
Zarech’e	*F*	*p* < 0.001	*p* = 0.009	*p* < 0.001
	*acc*	0.952	0.857	0.926
	*sens*	1.000	0.714	0.800
	*spec*	0.943	0.886	1.000
Sokolovo	*F*	*p* < 0.001	*p* < 0.001	*p* < 0.001
	*acc*	0.937	0.905	0.921
	*sens*	0.778	1.000	0.556
	*spec*	0.963	0.889	0.981
Lesnoye	*F*	*p* < 0.001	*p* = 0.141	*p* = 0.111
	*acc*	0.900	0.817	0.912
	*sens*	1.000	0.600	0.400
	*spec*	0.891	0.836	0.962

**Table 6 insects-12-00090-t006:** Results of linear modeling of tree-ring width indices. Results of linear modeling of tree-ring width indices. T is the mean temperature, and P is a monthly sum of precipitation. Lowercase month abbreviations represent previous year correlations with the weather, while uppercase represents the current year’s.

Model	Site	Selected Variables (Coefficients)						$R_{adj}^{2}$
Weather	Zarech’e	TJUL (−0.0189)	PJUN (0.0015)	Pjan (−0.0028)	Psep (0.0021)			0.395
	Sokolovo	Tjan (−0.0091)	PMAY (0.0017)					0.211
	Lesnoye	TJUL (−0.0232)	TAUG (−0.0279)	Psep (0.0018)	PMAY (0.0024)	PJUL (0.0018)		0.444
Weather/outbreak	Zarech’e	TJUL (−0.0163)	PJUN (0.0014)	Pjan (−0.0016)	Psep (0.0021)		O (−0.1879)	0.533
	Sokolovo	Tjan (−0.0086)	PMAY (0.0015)				O (−0.1871)	0.405
	Lesnoye	TAUG (−0.0202)	Psep (0.0020)	PMAY (0.0025)	PJUL (0.0019)		O (−0.1899)	0.502

## Data Availability

The data presented in this study are available on request from the corresponding author.

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
