# Peer review of "Pine Looper Bupalus piniaria (L.) Outbreaks Reconstruction: A Case Study for Southern Siberia"

_insects, 2021, doi:10.3390/insects12020090_

Round 1

Reviewer 1 Report

  • Line 30: What does “naturally determined” mean?
  • Line 33: Were there specific outbreaks that made you interested in the species?
  • Line 37: You cite forecasting as a applied
  • Line 59: Get rid of ‘by now’.
  • Line 63: List several of the negative influences.
  • Paragraph starting on line 68: This paragraph is very difficult to understand.
    • Does sentence #3 refer to individual pines that were not infested by the pine looper?
  • Sentence beginning on line 72: What is the name of the rarely used approach? Above, you stated that narrow tree rings can be caused by loopers, and other negative influences.  What are the tree ring characteristics that are specific to loopers?
  • Line 91: What is the base temperature for the growing degree day calculations?

Figure 1: Nice map!  You should add an arrow to make it clear that North is up.

Figure 2: love the idea of this figure, but it needs some adjustments to be effective:

  • There are too many vertical lines and arrows and they are all the same color. It’s difficult to visually discern the differences among:
    • Source-sink arrows
    • Between-year divisions
    • Early/late season divisions.
  • The inclusion of the tree ring image on the bottom is very effective.
    • I suggest extending the between-year lines through the ring image. This would make it easier to tell that the lines represent boundaries between growing years.
  • The patterns of the arrows connecting sources and sinks are largely redundant among years, for example.
    • Reserved carbohydrates are always sources for early wood growth.
    • Photosynthesis is always a source for new needle growth in the early season.
    • Photosynthesis is always a source for late wood growth and carbohydrate reserves in the late season
  • The large number of arrows clutters the visual space and makes it hard to differentiate the differently-sized circles. If you could eliminate the redundancy in the arrows, it would make your figure much cleaner and more readable. Perhaps you could include an extra column to serve as a guide that shows the flow directions.  Then you could omit the carbon flow arrows from the main part of the figure.
  • It’s not immediately obvious that the two columns within each year represent early and late growth. Maybe you could use a different color and/or texture for these lines?

  • Line 309: What is MCO?
  • Figures 3 – 5: The x-axes (year) are identical for all four methods within a site. I suggest uncluttering the figures by keeping the year axis on the lowest panel (MICA) and eliminating the redundant x-axis labels in the other three panels.  The figures appeared pixellated in the document that I received.  You should consider exporting your figures in a vector format like pdf.
  • Table 5: This information would be more impactful and easier to understand if it were presented as a figure instead of a table.
  • Line 366: What does ‘suddenly insufficient’ mean?
  • Line 455: It seems like the recovery pattern in Biysk and Minusinsk is the same?
  • Sentence starting on 455: does the sentence in this paragraph refer to Biysk, or Minusinsk? It’s unclear whether you are stating that defoliation was low in Biysk, in Minusinsk, or both.
  • Sentence starting on line 454: Do tress in Minusinsk have longer recovery times? It seems like that is the comparison you want to make, but it’s not explicitly stated.
  • Sentence starting on 472: This sentence fragment is incomplete and its meaning is unclear.
  • Do not use the terms ‘weather’ and ‘climate’ interchangeably. They are both important, but they are not equivalent!

Abstract and Introduction

The abstract has very little connection to the main points in the manuscript.  For example, it is not clear from the abstract that one of your objectives was to compare the performance of several reconstruction methods.  In the abstract, you prominently mention that the available time series data “are still not long enough”.  Upon reading the abstract, I expected the paper to deal with the implications of limited temporal scales in existing data, however that’s not a main focus of the actual work you for this paper.

In general, the introduction section contains a lot of relevant information, but it would greatly benefit from revisions for structure, streamlining redundant content, and English writing style. 

Additionally, there are many vague statements that don’t support the authors’ motivations.  For example, the second paragraph doesn’t give any specific reasons to motivate understanding the temporal dynamics of pine loopers.  It mentions “mechanisms of population dynamics” and “forecasting outbreaks”.  These general concepts could apply to many organisms. There is no connection to a larger motivation for studying pine looper without specific examples of why these are important in the context of the Scot’s pine/pine looper beetle system.

Methods Section

The methods section thoroughly describes the work the authors did.  In general, the English style was easier to understand than the Introduction section.

Discussion Section

You make many interesting and relevant points in the Discussion section, but the organization of this section needs to be improved.   For example, the first two subsections (4.1 and 4.2) are concise and flow logically from the section names, but the subsection 4.3 is extremely long and is hard to follow.  It would be much easier for the reader if you explicitly enumerated the possible factors contributing to mismatch you describe (different study systems/species, climate differences, defoliation intensity, etc.) at the beginning.  Then you can write about each of them in that order.  You should also more clearly differentiate the paragraphs so it’s clear which factor you are discussing.  I had to read through section 4.3 multiple times to understand the points you are trying to make.  It will be a very effective Discussion section if you can improve the organization!

Author Response

Dear colleague!

We are grateful to You for Your time and Your suggestions. We tried to follow Your comments when worked on manuscript. Here we submit the results.

Line 30: What does “naturally determined” mean?

Changed to: determined by intra-population processes (L115).

Line 33: Were there specific outbreaks that made you interested in the species?

To answer the question we have added: causes severe damage in pine forests [1,3] throughout Eurasia from North European Plain and Scotland to Transbaikalia, including Biysk pine forest (L117-118).

Line 37: You cite forecasting as a applied

Changed to: applied (prognosis of population dynamics and outbreaks) value (L51-52).

Line 59: Get rid of ‘by now’.

Removed from the last version of the manuscript.

Line 63: List several of the negative influences.

Listed in L92 (influenced by other negative impacts such as droughts [6] or forest fires [18]).

Paragraph starting on line 68: This paragraph is very difficult to understand.

Does sentence #3 refer to individual pines that were not infested by the pine looper?

The Introduction section have been rewritten completely, including this paragraph (see in attached file).

Sentence beginning on line 72: What is the name of the rarely used approach? Above, you stated that narrow tree rings can be caused by loopers, and other negative influences.  What are the tree ring characteristics that are specific to loopers?

These methods have been described in Reconstructions of outbreaks history section. Therefore we have added the references only. 

Line 91: What is the base temperature for the growing degree day calculations?

This information have been added to L136-137.

Figure 1: Nice map!  You should add an arrow to make it clear that North is up.

Added (see p. 4).

Figure 2: love the idea of this figure, but it needs some adjustments to be effective:

There are too many vertical lines and arrows and they are all the same color. It’s difficult to visually discern the differences among:

  • Source-sink arrows
  • Between-year divisions
  • Early/late season divisions.

The inclusion of the tree ring image on the bottom is very effective.

  • I suggest extending the between-year lines through the ring image. This would make it easier to tell that the lines represent boundaries between growing years.

The patterns of the arrows connecting sources and sinks are largely redundant among years, for example.

  • Reserved carbohydrates are always sources for early wood growth.
  • Photosynthesis is always a source for new needle growth in the early season.
  • Photosynthesis is always a source for late wood growth and carbohydrate reserves in the late season

The large number of arrows clutters the visual space and makes it hard to differentiate the differently-sized circles. If you could eliminate the redundancy in the arrows, it would make your figure much cleaner and more readable. Perhaps you could include an extra column to serve as a guide that shows the flow directions.  Then you could omit the carbon flow arrows from the main part of the figure.

It’s not immediately obvious that the two columns within each year represent early and late growth. Maybe you could use a different color and/or texture for these lines?

This figure have been redesigned according to Your suggestions.

Line 309: What is MCO?

I'm sorry, it's just a translation mistake. The MCO abbreviation changed to MWM (L395).

Figures 3 – 5: The x-axes (year) are identical for all four methods within a site. I suggest uncluttering the figures by keeping the year axis on the lowest panel (MICA) and eliminating the redundant x-axis labels in the other three panels.  The figures appeared pixellated in the document that I received.  You should consider exporting your figures in a vector format like pdf.

The x-axes of these figures have been redesigned (see pp. 11, 13, 14). 

It is sadly, but published .pdf figures looks sloppy and worse than in raster format, despite the latter give us the pixelated appearance.

Table 5: This information would be more impactful and easier to understand if it were presented as a figure instead of a table.

The table have been changed to figure (p. 17). 

Line 366: What does ‘suddenly insufficient’ mean?

The insufficient word changed to weak (L478).

Line 455: It seems like the recovery pattern in Biysk and Minusinsk is the same?

No, it is not. This section have been redesigned (see in attached manuscript), and I hope the ambiguity have been escaped. 

Sentence starting on 455: does the sentence in this paragraph refer to Biysk, or Minusinsk? It’s unclear whether you are stating that defoliation was low in Biysk, in Minusinsk, or both.

It refers to Biysk only. In revised manuscript this ambiguity have been eliminated (paragraph started on L556).

Sentence starting on line 454: Do tress in Minusinsk have longer recovery times? It seems like that is the comparison you want to make, but it’s not explicitly stated.

Yes, it does. This comparison is in L566-570.

Sentence starting on 472: This sentence fragment is incomplete and its meaning is unclear. 

This sentence have been rewrited (L589-591). 

Do not use the terms ‘weather’ and ‘climate’ interchangeably. They are both important, but they are not equivalent!

Currently this terms used by different ways. We use 'weather' when talking about conditions in the specific year(s) (L17, 557, 568, sections 3.5 and 4.3), but 'climate' when characterize a long-term situation on any territory (paragraph starting on L556).

Abstract and Introduction

The abstract has very little connection to the main points in the manuscript.  For example, it is not clear from the abstract that one of your objectives was to compare the performance of several reconstruction methods.  In the abstract, you prominently mention that the available time series data “are still not long enough”.  Upon reading the abstract, I expected the paper to deal with the implications of limited temporal scales in existing data, however that’s not a main focus of the actual work you for this paper.

Abstract have been completely rewritten.

In general, the introduction section contains a lot of relevant information, but it would greatly benefit from revisions for structure, streamlining redundant content, and English writing style. 

We tried to improve both structure and content of Introduction and rewrite it completely. In addition, we tried to made writing style better, but if our efforts are poor we 

Additionally, there are many vague statements that don’t support the authors’ motivations.  For example, the second paragraph doesn’t give any specific reasons to motivate understanding the temporal dynamics of pine loopers.  It mentions “mechanisms of population dynamics” and “forecasting outbreaks”.  These general concepts could apply to many organisms. There is no connection to a larger motivation for studying pine looper without specific examples of why these are important in the context of the Scot’s pine/pine looper beetle system.

We hope we did our motivation more obvious. With respect to Your remark, You can see the paragraphs starting on L57 and L73.

Methods Section

The methods section thoroughly describes the work the authors did. In general, the English style was easier to understand than the Introduction section.

Thank You for Your high evaluation. We add some new methods when was working on comments of other reviewers. 

Discussion Section

You make many interesting and relevant points in the Discussion section, but the organization of this section needs to be improved. For example, the first two subsections (4.1 and 4.2) are concise and flow logically from the section names, but the subsection 4.3 is extremely long and is hard to follow. It would be much easier for the reader if you explicitly enumerated the possible factors contributing to mismatch you describe (different study systems/species, climate differences, defoliation intensity, etc.) at the beginning.  Then you can write about each of them in that order. You should also more clearly differentiate the paragraphs so it’s clear which factor you are discussing. I had to read through section 4.3 multiple times to understand the points you are trying to make. It will be a very effective Discussion section if you can improve the organization!

This section have been rearranged according to Your advise (see from L556 to L606). 

Thanks again You for Your review. I'm hopeful that our changes of this text corresponds to Your suggestions.

Best regards!

Reviewer 2 Report

The manuscript of Demidko and colleagues, attempts a very interesting approach to reconstruct the outbreaks of Bupalus piniaria in Southern Siberia. To do that, the authors make use of dendrochronoloy, something that it is challenging on its own. In general the manuscript is efficiently written, and the authors employ a sufficient number of measurements to assess the outbreak history. In my opinion however, there are two main points that the authors need to carefully revise:

  1. the use of different approaches (PYM, MWM,  MICA and Outbreak), increases greatly the potential of the study. However, the authors should try to make the comparison of these approaches more easily accessible and conceivable to the reader, as in some cases it is difficult to fully undestand the point (f.e. Lines 421-436)
  2. for me it is always a big question, how credible is the association of tree-series to only one factor that might have influenced upon and largely determined them. In that sense, I would be more than grateful, if the authors provided some more details and information about how to exclude the possibility of other factors (biotic and/or abiotic) also affecting the tree ring series.

Minor points

Line 92: enouth for imago to emerge...

Lines 246-265: outof the nine minima, five of them correspond to outbreaks identified by PYM. What about the other minimum values? Do you have any idea what are they associated with?

Lines 330-331: This sentence needs to be revised, as here you try to bring a robust argument about the time lag, but in the way it is written, you do not make your point clear.

Lines 450-457: you need to pay additional attention to this paragraph. At the end of the paragraph you say that due to the low defoliation intensity "...accelerates the recovery of pre-outbreak growth rate". Here you mean either ""...accelerates the recovery of post-outbreak growth rate", or ""...accelerates the recovery growth rate at the pre-outbreak level". In general, I suggest that you elaborate more on this paragraph, in order to make it clearer. 

Author Response

Dear colleague!

We are grateful to You for Your time and Your suggestions. We tried to follow Your comments when worked on manuscript. Here we submit the results.

Some comments which do not require the answer are gray-colored.

The manuscript of Demidko and colleagues, attempts a very interesting approach to reconstruct the outbreaks of Bupalus piniaria in Southern Siberia. To do that, the authors make use of dendrochronoloy, something that it is challenging on its own. In general the manuscript is efficiently written, and the authors employ a sufficient number of measurements to assess the outbreak history. In my opinion however, there are two main points that the authors need to carefully revise:

The use of different approaches (PYM, MWM,  MICA and Outbreak), increases greatly the potential of the study. However, the authors should try to make the comparison of these approaches more easily accessible and conceivable to the reader, as in some cases it is difficult to fully undestand the point (f.e. Lines 421-436)

I'm sorry, but this long and in some degree complicated description is need for detailed analysis of obtained results. These sites looks similar, but statistical data shows some differences between them. For understanding of these differences and their impact on reconstruction methods we should discuss the results in details.

For me it is always a big question, how credible is the association of tree-series to only one factor that might have influenced upon and largely determined them. In that sense, I would be more than grateful, if the authors provided some more details and information about how to exclude the possibility of other factors (biotic and/or abiotic) also affecting the tree ring series.

Thank You very much! Really, it is a big disadvantage of previous version of this manuscript. We have used some procedures for splitting the impact of weather and defoliation (see sections 3.5 and 4.3 in attached file). We exclude the influence of forest fires, the second major factor of growth declining, because of there are not any evidences of fire damage in the studied site (see L147-148). We also don't find any symptoms of another disturbances (L146-147, L448-452).

Minor points

Thank You for Your corrections.

Line 92: enouth for imago to emerge...

Corrected (L137).

Lines 246-265: outof the nine minima, five of them correspond to outbreaks identified by PYM. What about the other minimum values? Do you have any idea what are they associated with?

Yes, we do. The results of our colleagues, who had investigated climate-growth relationships for this region, have been cited (L480-483).

Lines 330-331: This sentence needs to be revised, as here you try to bring a robust argument about the time lag, but in the way it is written, you do not make your point clear.

This point have been described more particularly in L433-435.

Lines 450-457: you need to pay additional attention to this paragraph. At the end of the paragraph you say that due to the low defoliation intensity "...accelerates the recovery of pre-outbreak growth rate". Here you mean either ""...accelerates the recovery of post-outbreak growth rate", or ""...accelerates the recovery growth rate at the pre-outbreak level". In general, I suggest that you elaborate more on this paragraph, in order to make it clearer. 

Thanks again You for Your review. I'm hopeful that our changes of this text corresponds to Your suggestions.

Best regards!

Reviewer 3 Report

This is a kind of innovative article, but some parts are inaccurate or I am not sure what they are. After reading this article, I have mixed feelings about whether it should be published on this stage. Based on the first part of the title, I expected an interesting article, but it turned out that I didn't find anything sure here, and the article is sometimes difficult to read and the main content is blurred. Moreover, the title may strongly indicate the local validity of these results. I suggest you to change the title to: An attempt to phyllophagous insects outbreaks reconstruction in Pinus sylvestris L.: a case study for Southern Siberia. In several places, you shyly admit that not only Bupalus piniaria (L.) could cause damage to the needles (L93: but also for others; L102: “and/or”; L128-129 etc.). Therefore, I suggest that you do not emphasize this species at every step. The conclusions of the manuscript aren’t well supported by the results of research obtained in a study.

The article concerns an important issue of reconstruction of outbreaks in Pinus sylvestris L. using dendrochronological method. The biggest disadvantage is the introduction, which does not properly introduce to the content of the paper. It does not show exactly what we already know and what is new here, and why it is worth researching. The paper lacks a clearly defined research goals and no hypothesis has been made at the end of the manuscript.

Specific comments:

  1. I am not a native speaker, but sometimes the English language seems to need additional checking. For example, there are some strange or incomprehensible collocations, too long sentences, etc. (eg. L21: „literaty sources data” instead „literature sources data”).

  1. In addition, the results (especially figures) look good and well prepared. The discussion is a little better than introduction, but the ratio of discussion length to conclusion is inadequate. I believe that the discussion is so extensive and is unnecessary. I suggest reviewing it and try to shorten it as much as possible (for example look at L371-387).

  1. The description of chapter 2.4 is quite difficult and it is easy to get lost. Perhaps it would be worth presenting it in a more accessible form, for example adding a graphic abstract of these methods.

  1. There are many different shortcuts here. Take a look at tables 3 and 4 for example. Wouldn't it be better to add a legend?

  1. Conclusions require very large changes. Here we expect generalizations and observations, not repetitions or one substantive sentence, and I suppose it is incorrect (L498-500). Is the end of the sentence correct? As it corresponds to Table 5... Perhaps it is worth adding a few specific sentences here so as not to embarrass the reader. You also wrote little in whole paper about the historical data. Are we sure about them? Maybe they are faulty?

  1. You don't care about the details and the editorial side of your work:

No italics: L25, L149, L286, L343, L590

L130 Don't start a sentence with an abbreviation.

L360-361 Bad citation style. Will the numbering match now?

L468 P. menizesii instead of full name (look at L385)

L522 Scots

Author Response

Dear colleague!

We are grateful to You for Your time and Your suggestions. We tried to follow Your comments when worked on manuscript. Here we submit the results.

Some comments which do not require the answer are gray-colored.

Comments and Suggestions for Authors

This is a kind of innovative article, but some parts are inaccurate or I am not sure what they are. After reading this article, I have mixed feelings about whether it should be published on this stage. Based on the first part of the title, I expected an interesting article, but it turned out that I didn't find anything sure here, and the article is sometimes difficult to read and the main content is blurred. Moreover, the title may strongly indicate the local validity of these results. I suggest you to change the title to: An attempt to phyllophagous insects outbreaks reconstruction in Pinus sylvestris L.: a case study for Southern Siberia. In several places, you shyly admit that not only Bupalus piniaria (L.) could cause damage to the needles (L93: but also for others; L102: “and/or”; L128-129 etc.). Therefore, I suggest that you do not emphasize this species at every step. The conclusions of the manuscript aren’t well supported by the results of research obtained in a study.

The article concerns an important issue of reconstruction of outbreaks in Pinus sylvestris L. using dendrochronological method. The biggest disadvantage is the introduction, which does not properly introduce to the content of the paper. It does not show exactly what we already know and what is new here, and why it is worth researching. The paper lacks a clearly defined research goals and no hypothesis has been made at the end of the manuscript.

The title have been changed. 

The sentence from L102 (L149 after revision) have been slightly rewritten. To avoid this ambiguity in other parts of manuscript we have added this sentence to L118-119: 'including Biysk pine forest where the pine looper is the only mass Scots pine defoliator'.

The Introduction and Conclusion sections have been rewritten completely (see the revised version).

Specific comments:

I am not a native speaker, but sometimes the English language seems to need additional checking. For example, there are some strange or incomprehensible collocations, too long sentences, etc. (eg. L21: „literaty sources data” instead „literature sources data”).

We have proof-readed this text one more time. If it is not enough, we will use MDPI Author Editing Service. 

In addition, the results (especially figures) look good and well prepared. The discussion is a little better than introduction, but the ratio of discussion length to conclusion is inadequate. I believe that the discussion is so extensive and is unnecessary. I suggest reviewing it and try to shorten it as much as possible (for example look at L371-387).

I agree, 5.1 (4.1 before revision) section is a little bit longer that possible. But it contains a lot of details which are necessary for good interpretation. Sadly, but visually similar sites gave us not quite overlapping results. Therefore, we should discuss these results separately and in details.

The description of chapter 2.4 is quite difficult and it is easy to get lost. Perhaps it would be worth presenting it in a more accessible form, for example adding a graphic abstract of these methods.

The comprehensive description of used methods is necessary for better understanding of their outcomes. I'm afraid, text reduction can make hard the reading of Discussion section. 

There are many different shortcuts here. Take a look at tables 3 and 4 for example. Wouldn't it be better to add a legend?

I try to merge the data from these tables with both legends and figures. I'm sorry, but this procedure made them very overloaded and confusing.

Conclusions require very large changes. Here we expect generalizations and observations, not repetitions or one substantive sentence, and I suppose it is incorrect (L498-500). Is the end of the sentence correct? As it corresponds to Table 5... Perhaps it is worth adding a few specific sentences here so as not to embarrass the reader. You also wrote little in whole paper about the historical data. Are we sure about them? Maybe they are faulty?

It was our big mistake to wrote so brief Conclusions. This section have been rewritten completely for better compliance with goals and results (see in attached manuscript). 

You don't care about the details and the editorial side of your work:

All these imperfections have been corrected (see further).

No italics: L25, L149, L286, L343, L590

L45, L195, L372, L454, L757

L130 Don't start a sentence with an abbreviation.

L120

L360-361 Bad citation style. Will the numbering match now?

L472

L468 P. menizesii instead of full name (look at L385)

L498-499

L522 Scots

L640

Thanks again You for Your review. I'm hopeful that our changes of this text corresponds to Your suggestions.

Best regards!

Reviewer 4 Report

General comments

The manuscript entitled “Pine looper Bupalus piniaria (L.) outbreaks reconstruction: a case study for Southern Siberia” ID: 996673 explore the effectiveness of different methods to identify B. piniaria outbreaks and reconstruct the time series of the events in the Altai Krai area. The methods are based on specific patterns encoded in the early wood and late wood.  

Several issues need to be addressed before considering this manuscript suitable for publication.

The aim, the novelty, the timing and the importance of this study need to be clearly addressed. These aspects should be highlight throughout the manuscript. Why is so important to reconstruct the outbreak events of B. piniaria (L.) in the area? Is it because due to climate change these defoliations events are expected to increase in frequency and intensity in the near future, compromising the pine stand that are of economically interest for the area? Please consider to add eventually some hypothesis.

The abstract is vague and method vs methods result confusing.

The background information and the gap of knowledge are not fully addressed.

The methods used are incomplete.

Not clear if the manuscript conclusion are supported by the results as the different methods show a weak correspondence between them.

Not clear how the authors have disentangle the key drivers (i.e. climate or insects) of narrow ring formation.

Literature review is pertinent but could be definitely integrated (see specific comments below).

The text would benefit from an English proofreading.

Specific comments

Abstract

L12-15 Please be more explicit

L15-17 Missing verb.

L17-21 Method, methods or analytical methods?

L20-22 and what about the climate?

Why is so important to reconstruct the time series of B. piniaria abundance?

Introduction

Please contextualize the entire work.

L38 Reference is missing.

L43 What do the authors mean with “only outbreak data”? Which kind of data are the authors referring to?

L49 Is the economic importance of the specie the most important reason for this study?

L56 Please replace “damaged” with “affected”.

L58-59 Please rephrase to improve clarity.

L61-64 Please be more explicit.

L66-67 See Babst et al., 2010

L68-71 Several papers have already published different methods to disentangle the key drivers of narrow ring formation (see Wilmking et al. 2018, Babst et al., 2010, Peters et al. 2017, 2018, Castagneri et al., 2019).

L74-79 Please state which methods have been used.

Methods

L82 “Samples have been collected in pine stands of the lower…”

L86-89 Please rephrase as e.g. “The study area is located on the territory of the…that are classified as forest-steppe”.

L88 Flat land?

L89-91 Is the are affected by continental climate?

L98 This has already been stated at L 82-83

L99-100 Are these stands characterized by anthropogenic disturbances?

L104 Be more explicit.

L105 the 3 sites are mentioned for the 1st time in the table 1. This should have been clarified before.

L107 How many trees have been sampled? How many cores per trees? Was compression wood avoided? Did you follow the standard protocol? Did you sand the cores with fine grain papers before scanning the images?

L111 Does this detrending method have a name? Please add references.

L113-115 Please rephrase. Not clear why the tested series have been excluded.

L117 “too small” is too vague? Is there any quantitative threshold?

L120 …and calculated using dplR package.

L122 Not clear how many samples have been discharged.

L123 You could think about splitting this chapter considering dedicating an entire paragraph to the specie background as in Lund et al., 2017.

L129 Therefore, the damages are expected to occur only in the latewood. Is it correct?

L138-139 Please clarify how the 1st order autocorrelation indirectly indicate the active use of accumulated reserves.

L140 Red maple trees are angiosperms. Is the cited study valid also for evergreen conifers?

L148 Please take into account that pine is an evergreen conifer.

L160 references are missing (see Kaennel & Schweingruber, 1995, Cropper, 1979; Neuwirth, Schweingruber, & Winiger, 2007) but see also pointRes package (van der Maaten- Theunissen, Maaten, & Bouriaud, 2015). In general, please be consistent with the specific terminology used in previous literature.

L172 Not clear if your trees belong to this case. See also Cropper, 1979; Neuwirth, Schweingruber, & Winiger, 2007

L183 Please be consistent with the specific terminology used in previous literature.

L184-184 Please clarify this sentence.

L198 Lim and Ing have not been introduced before.

L207-212 It is still not clear how the authors disentangle the effect of the biotic disturbance from the climate signal.

L230 Why has PointRes package not been used/considered (see van der Maaten- Theunissen, Maaten, & Bouriaud, 2015)?

Results

L278-283 It is not clear if the effect of outbreaks events on growth is synchronised with the event, lagged and in general how many years are needed before the full recovery. You could run a superposed epoch analysis (Holmes & Swetnam, 1994; Swetnam & Betancourt, 1990) to investigate this aspect see peters et al., 2017

L290 What does relative reliable mean?

L290+L310+L312+L326-327+L349-351+L403 It seems the authors did not find a strong correspondence between the results of the different analytical methods used.

L339 Please quantify “much”

L357 Please remove “did”.

L359 What does “sufficient” mean?

L408 “..the signal detected by this method…”

L435 Add references.

L465-466 This aspect could be tested running climate growth relationship analysis in the area.

L468 “Franco”?

L472 Missing references.

L487 “viable”?

L490-492 This sentence seems to be in contrast with your results. In addition, references are missing.

L496-500 Method, methods, or analytical methods? Please clarify the conclusion, as it seems to be in contrast with the results (see also comment before L290…).

Author Response

Dear colleague!

We are grateful to You for Your time and Your suggestions. We tried to follow Your comments when worked on manuscript. Here we submit the results.

Some comments which do not require the answer are gray-colored.

General comments

The manuscript entitled “Pine looper Bupalus piniaria (L.) outbreaks reconstruction: a case study for Southern Siberia” ID: 996673 explore the effectiveness of different methods to identify B. piniaria outbreaks and reconstruct the time series of the events in the Altai Krai area. The methods are based on specific patterns encoded in the early wood and late wood.  

Several issues need to be addressed before considering this manuscript suitable for publication.

The aim, the novelty, the timing and the importance of this study need to be clearly addressed. These aspects should be highlight throughout the manuscript. Why is so important to reconstruct the outbreak events of B. piniaria (L.) in the area? Is it because due to climate change these defoliations events are expected to increase in frequency and intensity in the near future, compromising the pine stand that are of economically interest for the area? Please consider to add eventually some hypothesis.

The abstract is vague and method vs methods result confusing.

The background information and the gap of knowledge are not fully addressed.

We try to correct the main disadvantages (underlined) You pointed out. Abstract and Introduction sections have been rewritten completely for this purpose.

The methods used are incomplete.

In order to make our results more reasonable we used additionally linear modelling (for split the weather impact from the defoliation one) and SEA. For more detailed view, see 3.5, 3.6 and 4.3 sections.

Not clear if the manuscript conclusion are supported by the results as the different methods show a weak correspondence between them.

In part, I agree with You. Actually, used methods are not completely correspondent to each other. But these methods have got different specifity and sensitivity, and we can't expect that their results will be totally coinciding. That's why we use a set of such methods (also, see Paritsis et al. Can. J. For. Res., 2009; Fan, Bräuning, Ecological Indicators, 2017; Büntgen et al. New Phytol., 2009).

Not clear how the authors have disentangle the key drivers (i.e. climate or insects) of narrow ring formation.

Thank You very much for references to Wilmking et al. and Peters et al. It was very useful for this manuscript. We used linear modelling for splitting wether effect and insect one (3.5 and 4.3 sections). The second main driver of narrow ring formation, severe fires, was absent in the studied area (L147).

Literature review is pertinent but could be definitely integrated (see specific comments below).

The text would benefit from an English proofreading.

The language correction have been performed one more time. If it is not enough, we are planning take the opportunity to use MDPI Author Editing Serivce.

Specific comments

Abstract

L12-15 Please be more explicit

L15-17 Missing verb.

L17-21 Method, methods or analytical methods?

L20-22 and what about the climate?

Why is so important to reconstruct the time series of B. piniaria abundance?

The Abstract have been completely rewritten. 

Introduction

The Introduction have been completely rewritten, but some comments require our response. 

Please contextualize the entire work.

We have tried to do it (see the whole section).

L38 Reference is missing.

The reference have been added (L52). 

L43 What do the authors mean with “only outbreak data”? Which kind of data are the authors referring to?

The sentence 'only outbreak data' have been changed to 'only defoliation records' (L56). Also, we add two references with examples of this approach. 

L49 Is the economic importance of the specie the most important reason for this study?

No, it is not. I guess, our motivation became clearer in the last version of Introduction

L56 Please replace “damaged” with “affected”.

It's done (L75).

L58-59 Please rephrase to improve clarity.

It's done (L78-80).

L61-64 Please be more explicit.

These sentences have been completely removed.

L66-67 See Babst et al., 2010

I had found only this article: 'Age and susceptibility of Fennoscandian mountain birch (Betula pubescens) towards insect outbreaks'. But it is unrelated to past outbreak reconstruction.

L68-71 Several papers have already published different methods to disentangle the key drivers of narrow ring formation (see Wilmking et al. 2018, Babst et al., 2010, Peters et al. 2017, 2018, Castagneri et al., 2019).

Thank You for these references! We have processed our data for disentangling ring formation drivers (see sections 3.5 and 4.3).  

L74-79 Please state which methods have been used.

These methods are described in details in 3.4 section, thus we just add references to articles where the methods had been used (L90-91).

Methods

L82 “Samples have been collected in pine stands of the lower…”

Thank You very much. This correction have been made (L127).

L86-89 Please rephrase as e.g. “The study area is located on the territory of the…that are classified as forest-steppe”.

It's rewritten (L131-132).

L88 Flat land?

Yes, it is (see L132-133).

L89-91 Is the are affected by continental climate?

No, it is not. Such values of temperature and precipitation are optimal (at least, suboptimal)  for both pine and looper. 

L98 This has already been stated at L 82-83

It have been corrected (L127 and L144-145).

L99-100 Are these stands characterized by anthropogenic disturbances?

The clarification have been added (L145-147).

L104 Be more explicit.

See L149-150.

L105 the 3 sites are mentioned for the 1st time in the table 1. This should have been clarified before.

All the sites have been mentioned in L145 and L147. 

L107 How many trees have been sampled? How many cores per trees? Was compression wood avoided? Did you follow the standard protocol? Did you sand the cores with fine grain papers before scanning the images?

A number of trees and cores specified in Table 1.

A clarification concerning compression wood have been made in L154.

Standard protocols (St. Andrew’s protocol and further) had been designed for blue intensity measurement, but not for ring-width or seasonwood. In the case of such kind of measurements authors mentioned about preparing&scanning procedures and used software (for example, Rydval et al. Can. J. For. Res., 2015).

The sanding procedure have been described in L153-154.

L111 Does this detrending method have a name? Please add references.

This method is embedded in CDendro software. I have added a reference to it.

L113-115 Please rephrase. Not clear why the tested series have been excluded.

Of course, it need some more details. I have used the widespread ‘leave-one-out’ approach. It mentioned consecutive exclusion of tested objects from sample for cross-validation. It need for testing only; if correlation coefficient of tested series and site chronology (calculated after removing tested series) is more than threshold value, the series put back to sample. I have mentioned the ‘leave-one-out’ method in the text (L164) for clarifying.

L117 “too small” is too vague? Is there any quantitative threshold?

The quantitative threshold have been added to L166.

L120 …and calculated using dplR package.

It is corrected (L169-170).

L122 Not clear how many samples have been discharged.

It is added to 4.1 section (L322).

L123 You could think about splitting this chapter considering dedicating an entire paragraph to the specie background as in Lund et al., 2017.

Thank You very much for this advice. We have add this section (see in text).

L129 Therefore, the damages are expected to occur only in the latewood. Is it correct?

No, it is not. Pine looper defoliation have got persistent effect (see Figure 1 and results of superposed epoch analysis in L328-329).

L138-139 Please clarify how the 1st order autocorrelation indirectly indicate the active use of accumulated reserves.

This clarification have been added to L184-185.

L140 Red maple trees are angiosperms. Is the cited study valid also for evergreen conifers?

I agree, it is not quite reliable analogy. Sadly, only small number of such articles are occur, and I forced to use any available literature source.

L148 Please take into account that pine is an evergreen conifer.

This effect is known for evergreen species too (L193). 

L160 references are missing (see Kaennel & Schweingruber, 1995, Cropper, 1979; Neuwirth, Schweingruber, & Winiger, 2007) but see also pointRes package (van der Maaten- Theunissen, Maaten, & Bouriaud, 2015). In general, please be consistent with the specific terminology used in previous literature.

We have proposed and described this method firstly. There are not any references for it. 

The functionality of pointRes are excessive for our goal. 

We use 'pointer year' term in accordance with Becker et al. Ann. Sci. For., 1994, as in the dplR package.

L172 Not clear if your trees belong to this case. See also Cropper, 1979; Neuwirth, Schweingruber, & Winiger, 2007

The lack of suitable tree species for building control undamaged tree-ring series have been mentioned in L86-88. Apart from Scots pine, the prevalent tree species of study territory are birch and (in artificial stands) poplar. Both these species are less rot-resistant and long-lived compared to pine.

Both Cropper and Neuwirth & colleagues wrote about methods of pointer years identification, but not about past outbreak detection. It is known that narrow rings decection can be used for this purpose as auxiliary only (Paritsis et al. Can. J. of For. Res., 2009; Fan, Bräuning. Ecol. Indicators, 2017).

L183 Please be consistent with the specific terminology used in previous literature.

This term is consistent with earlier articles. For example (Büntgen et al. New Phytologist, 2009): ‘Six methods were applied … (iii) a simple 15-yr moving window approach’.

L184-184 Please clarify this sentence.

This sentence have been improved (L232-235).

L198 Lim and Ing have not been introduced before.

The explanation have been added to L248.

L207-212 It is still not clear how the authors disentangle the effect of the biotic disturbance from the climate signal.

See sections 3.5 about methods and 4.3 about results. 

L230 Why has PointRes package not been used/considered (see van der Maaten- Theunissen, Maaten, & Bouriaud, 2015)?

The procedures of pointer years identification from pointRes package are more complex than Becker's and colleagues algorithm. Possibly, it can give some advantages, but for first time we prefer use simpler approach.

Results

L278-283 It is not clear if the effect of outbreaks events on growth is synchronised with the event, lagged and in general how many years are needed before the full recovery. You could run a superposed epoch analysis (Holmes & Swetnam, 1994; Swetnam & Betancourt, 1990) to investigate this aspect see peters et al., 2017

It is done (L305, L328). Thank You very much!

L290 What does relative reliable mean?

It have been rewritten (L376). 

L290+L310+L312+L326-327+L349-351+L403 It seems the authors did not find a strong correspondence between the results of the different analytical methods used.

Yes, it is. It is due to low sensitivity of some methods (independent component analysis and, in some cases, mowing window) or due to lags between used methods (see 4.2 section for details). Also there is possibility of lags between reconstruction and Forest Service information as described in (Speer et al. Ecology, 2001).

L339 Please quantify “much”

I’ve changed ‘much’ by qualitative description (L449-450).

L357 Please remove “did”.

Removed.

L359 What does “sufficient” mean?

It have been specified: 'sufficient reliability level in terms of statistical comparison of their results' (L471).

L408 “..the signal detected by this method…”

Proofed (L522).

L435 Add references.

References have been added (L549).

L465-466 This aspect could be tested running climate growth relationship analysis in the area.

We imply not weather conditions in the specific year but probability of long (multi-year) impact of unfavorable weather (in L557 'weather' have been changed to 'climatic' for unambiguity), which can impede foliage recovery. Of course, radial increment in studied forest depends on climate in some degree, but harsh weather is rare there (see paragraph starting from L556).

L468 “Franco”?

Yes; see, for example, https://www.gbif.org/species/2685796.

L472 Missing references.

The reference have been added (L589).

L487 “viable”?

In (Baltensweiler, Fishlin, 1988) there is this statement: ‘The result is that many needles are only nibbled or partly consumed’. I agree it is not obvious that such needles are alive; this claim have been removed.

L490-492 This sentence seems to be in contrast with your results. In addition, references are missing.

The necessary clarification and reference have been added to L605. 

L496-500 Method, methods, or analytical methods? Please clarify the conclusion, as it seems to be in contrast with the results (see also comment before L290…).

The Conclusions have been rewritten completely with regards to this Your comment (see in text).

Thanks again You for Your review. I'm hopeful that our changes of this text corresponds to Your suggestions.

Best regards!

Round 2

Reviewer 3 Report

Dear Authors,

I am sorry you have waited so long for my reply. I wanted to read your work thoroughly once again. Thank you very much for that you used my suggestions. In addition, I noticed that you responded to comments from other reviewers. At this stage, I do not see any contraindications for the MS to be accepted. Good luck with your further research and in the New Year.

Author Response

Dear collegue, 

I appreciate You for Your time and efforts. Your advices and comments was really helpful for us. 

Best regards, 

Denis Demidko and co-authors

Reviewer 4 Report

The current version of the manuscript entitled “Pine looper Bupalus piniaria (L.) outbreaks reconstruction: a case study for Southern Siberia” ID: 996673 has improved compared to the previous version.

Despite this, there are still several issues that need to be addressed before considering this manuscript suitable for publication.

Methods, results and discussion are still confusing.

The link between paragraphs in the discussion could be improve to guide the reader thought it.

There are still uncertainties in the methods that need to be addressed. In addition, they could be shortened.

The text would benefit from an English proofreading.

See also the specific comments below.

Simple summary

L13-14 Please rephrase “one has to resort…”

L18 Please rephrase “What is more, it is more…”

Abstract

L24-26 This is not true. There are numerous papers documenting methods to reconstruct outbreaks dynamics (e.g. see the literature cited by  Rolland and Lempérière 2004).

L27 This is not totally true as non-infested sites with the same tree species as a non-host reference could also be used to reconstruct outbreaks dynamics (Weber and Schweingruber, 1995).

L32 Not clear which are the disadvantages mentioned here.

L34-36 Is this valid for all the individuals?

L37-38 Please rephrase. Are the response significantly different?

40-44 Could you generalize your findings in the way of highlighting the importance of your work?

Introduction

L58 Not clear if it is the phyllophagous population of economic interest or the host tree species.

L68-71 Please split the sentence to improve clarity.

L69 Where is Biysk located? As the explanation comes only at L 131-133.

L71-72 Please rephrase “…saving us the trouble”…

L73 Please rephrase “one can only reconstruct…” with “ pest outbreaks history could only be reconstruct to a varying degrees of reliability”.

L78 see also the combination of methods used in Babst et al., 2010 to assess the spatiotemporal patterns of the autumnal moth (Epirrita autumnata) in northernmost Fennoscandia.

L80 I suggest to add “central” before “Asia”. See “Most of the work using dendrochronology in Asia was by Carus, who produced four publications on defoliators in Turkey (Carus 2004, 2009, 2010; Carus and Avci 2005) with much of his work focusing on pine processionary moth (Thaumetopoea wilkinsoni). Other important work has been conducted in Russia, Georgia, Mongolia, and India. Sarajishvili (1997) studied new pine knot-horn moth in Georgia near the Black Sea, Sviderskaya and Pal’nikova (2003) studied pine looper (Bupalus piniarius) in Russia, and Kucherov (1991) studied gypsy moth (Lymantria dispar) in the Ural Mountains. Studies in central Asia include those in Mongolia (Dulamsuren et al. 2010) on gypsy moth and in India (Priya and Bhat 1998) on insect damage in teak (Tectona grandis). Whereas Turkey and Russia have been well studied, much more work remains to be done in Asia, especially in the central region of the continent”. From Speer and Kulakowski 2017.

L84 This is not the only way as e.g. non infested sites with the same tree species as a non-host reference could also be used to reconstruct outbreaks dynamics (Weber and Schweingruber, 1995).

L92 Ring width could be negative influenced by other factor such as drought or forest fires.

L93 Please rephrase “One can solve…”

L97 Probably the LW would not only become narrower but also less dense (lighter) due to the defoliation.

L101 The efficiency of the methods should be tested.

L107-110 Please rephrase.

Methods

L137 “Imago”?

L156-158 Please rephrase. How many samples have been excluded from the analysis? How many missing rings have been identified?

L159 Please add the reference that correspond to the method used not about the software that have been used.

L166 remove “small”

L170 This is a normal procedure, the measurement of several radii are averaged to one series per tree. I would identified it as a tree series and not tree chronology. Normally a chronology belongs to a site or a sub site (and t is based on several tree series).

L179-194 It seams that this paragraph could fit better to the introduction, for explaining the reason of choosing this new method.

L196 Remove “by”.

L198 Have been radial growth consider yearly? Or EW and LW were separated when calculating the TGC?

L197-204 Here the authors should clarify if EW and LW have been consider independently or not. Then the formula should be adjusted to clarify this point and show how EW and LW width have been considered/used.

L209 The scheme in Fig. 2 does not correspond to what is written at L 34-36.

L215 Please replace “damage” with “affected”. Same comment L219.

L218 Please add reference.

L222 Spline detrending is different from the one mentioned at L159 and L265. Which standardization method did you use in this study? Did you check the stability of the results using different detrending methods? The use of different standardization in different methods could explain the fact that results do not fully correspond.

L227 The object of the sentence is missing or you should remove “to”.

L227-231 Could you please clarify this description.

L245 Does independent component fast analysis correspond to MICA?

L263 Consider to replace “weather” with “climate” as temperature and precipitation are generally considered climate data.

L276 It’s not clear how the disturbance factor has been included into the model. What do you mean for a dummy variable? Do you mean categorical?  How did you account for intraspecific variability of responses? E.g. using random factor in Linear mixed effect (LME) models? You could disentangle between the climate and outbreak effect considering both numerical (climate) and categorical (outbreak) variable in the models. In addition, it is not clear if EW and LW width  have been considered separately or not.

L287 after defoliation/crown reduction.

L323 Replace “Basing” with “based”.

Results

L329 Please link to table or figure with the results. What do “optimal parameter values” mean?

L404 Is the correlation significant?

L406 Radial growth also correlate with… did you also test EW and LW separately?

L409 See also Wilmking et al., 2018, Prendin et al., 2020.

Discussion

The discussion should take into account the results from the LME models.

L555 Here climate (and not weather as before) is mentioned. Is this a subchapter? “The 1st possible reason” of what?

L573, 592, 596 Is it another subchapter?

Author Response

Dear collegue! 

We are grateful to You for one more round of advices, and we hope Your efforts are not lost.

Methods, results and discussion are still confusing.

&

The link between paragraphs in the discussion could be improve to guide the reader thought it.

&

There are still uncertainties in the methods that need to be addressed. In addition, they could be shortened.

We have reworked the structure and, partly, the text of these sections. I hope, they became better after this revision.

L24-26 This is not true. There are numerous papers documenting methods to reconstruct outbreaks dynamics (e.g. see the literature cited by Rolland and Lempérière 2004).

I afraid, we misunderstand each other. In this sentence, I mentioned direct observations but not methods of reconstruction. Indeed, regular outbreak monitoring in former USSR had started about mid-XX century, in USA, as far as I know, about 1920-1930. In addition, the records might not have been saved. However, by dendrochronological methods we can create more long-lasting chronology of outbreaks (for duration of documented and reconstructed outbreaks history see, for example, fig. 3 in: Speer et al. Ecology, 2001: 82(3)).

L32 Not clear which are the disadvantages mentioned here.

It is impossible to talk about the disadvantage detailed in the abstract, then I had added an example of such things '(e.g., weak specificity)' (L32; here and after line numbers are given for .docx; I afraid, the shifts are possible in .pdf version).

L34-36 Is this valid for all the individuals?

This is not obligate. Moreover, it is unwanted approach, because of it can hide weak outbreaks: 'However, outbreaks that are visible in only a few trees may be averaged out of mean ring-width series' (Weber and Schweingruber, 1995). For clarification, I have added in this sentence '…among the majority of individuals' (L36).

L37-38 Please rephrase. Are the response significantly different?

Rephrased (L36–38).

40-44 Could you generalize your findings in the way of highlighting the importance of your work?

Thank You. I've rephrased this part of text (L40–45).

L58 Not clear if it is the phyllophagous population of economic interest or the host tree species.

Thank You very much; this ambiguity have been corrected (L60–62).

L68-71 Please split the sentence to improve clarity.

&

L69 Where is Biysk located? As the explanation comes only at L 131-133.

Done: '…Biysk pine forest (southeastern part of WSP's forest-steppe) was not infested by these species' (L69-73).

L73 Please rephrase “one can only reconstruct…” with “ pest outbreaks history could only be reconstruct to a varying degrees of reliability”.

Thank You very much. It is done (L75–76).

L80 I suggest to add “central” before “Asia”. See “Most of the work using dendrochronology in Asia was by Carus, who produced four publications on defoliators in Turkey (Carus 2004, 2009, 2010; Carus and Avci 2005) with much of his work focusing on pine processionary moth (Thaumetopoea wilkinsoni). Other important work has been conducted in Russia, Georgia, Mongolia, and India. Sarajishvili (1997) studied new pine knot-horn moth in Georgia near the Black Sea, Sviderskaya and Pal’nikova (2003) studied pine looper (Bupalus piniarius) in Russia, and Kucherov (1991) studied gypsy moth (Lymantria dispar) in the Ural Mountains. Studies in central Asia include those in Mongolia (Dulamsuren et al. 2010) on gypsy moth and in India (Priya and Bhat 1998) on insect damage in teak (Tectona grandis). Whereas Turkey and Russia have been well studied, much more work remains to be done in Asia, especially in the central region of the continent”. From Speer and Kulakowski 2017.

Thank You very much. It's done (L82). Special thanks for Turkish, Indian and Georgian references.

L27 This is not totally true as non-infested sites with the same tree species as a non-host reference could also be used to reconstruct outbreaks dynamics (Weber and Schweingruber, 1995).

&

L84 This is not the only way as e.g. non infested sites with the same tree species as a non-host reference could also be used to reconstruct outbreaks dynamics (Weber and Schweingruber, 1995).

I agree, but this way is risky. Such sites could have been affected before onset of documented period. See, for example, citation about the same situation: 'This site [predominated by host tree species], however, was selected based on digital change detection [available since 1986 and later] and is therefore only guaranteed to be outbreak-free back to 1986' (Babst et al. Remote Sensing of Environment, 2010: 114).

L92 Ring width could be negative influenced by other factor such as drought or forest fires.

Thank You very much. The sentence have been rephrased (L93–95).

L97 Probably the LW would not only become narrower but also less dense (lighter) due to the defoliation.

I'm in complete agreement with You. Moreover, this effect has been used for outbreak detection (e.g., Paritsis et al. Can. J. For. Res., 2009: 39). However, the difference between normal and light rings can be uncertain. By this reason, we deliberately avoided this feature of the past outbreaks. It is possibly that good results can be obtained by blue intensity, but by this moment, we have not finished the work.

L101 The efficiency of the methods should be tested.

Thank You; it is done (L120–121).

L137 “Imago”?

For better clarity, this sentence have been rephrased (L155–156), information about growth degree-days have been relocated to 'Species background' section.

L156-158 How many samples have been excluded from the analysis?

&

L156-158 How many missing rings have been identified?

See in 'Tree-ring series characteristics and search for the optimal parameter values' subsection (L311–312).

L159 Please add the reference that correspond to the method used not about the software that have been used.

This method changed to well-described Baillie & Pilcher method also implemented to CDendro (L179) (with the same results). The reference has been added.

L166 remove “small”

Thank You; it's done (L181).

L170 This is a normal procedure, the measurement of several radii are averaged to one series per tree. I would identified it as a tree series and not tree chronology. Normally a chronology belongs to a site or a sub site (and t is based on several tree series).

Thank You very much. Of course, it should be used 'series' but not 'chronology'. This sentence has been corrected (L181–183).

L179-194 It seams that this paragraph could fit better to the introduction, for explaining the reason of choosing this new method.

This paragraph have been relocated (see Introduction).

L196 Remove “by”.

It's done (L197).

L198 Have been radial growth consider yearly? Or EW and LW were separated when calculating the TGC?

For clarity, I have added '(separately for early- and latewood)' (L199–200).

L197-204 Here the authors should clarify if EW and LW have been consider independently or not.

I hope, this correction: searching for pointer years … in tree-ring series built for early- and latewood 'simultaneously ' (L197) is enough for better clarity.

L197-204 Then the formula should be adjusted to clarify this point and show how EW and LW width have been considered/used.

This outbreak detection algorithm is hard to be displayed as a formula. We have used the form like in (Swetnam et al. USDA Forest Service PNW-RP-484, 1995) for OUTBREAK algorithm. Also, we update the Fig. 2 for better explanation of this algorithm.

L209 The scheme in Fig. 2 does not correspond to what is written at L 34-36.

'The past defoliation [in year t] by the pine looper is indicated by the presence of a negative pointer year for latewood [in year t+1], followed by a negative pointer year for earlywood in a subsequent [t+2] year'.

On the scheme, after defoliation in year t latewood width declines abruptly (~3-fold) in the next t+1 year. Then, in t+2 year the earlywood width declines by >10%. It is in consistence with cited fragment of Abstract and threshold values from Table 3.

L215 Please replace “damage” with “affected”. Same comment L219.

Thank You; it's done (L215 and L220).

L218 Please add reference.

The reference have been added (L218).

L222 Spline detrending is different from the one mentioned at L159 and L265. Which standardization method did you use in this study? Did you check the stability of the results using different detrending methods? The use of different standardization in different methods could explain the fact that results do not fully correspond.

We use double detrending method for weather-growth relationships investigations only. The advantages of this method briefly described in Methods of Dendrochronology… (1992). Also, Baillie & Pilcher normalization method has been used for crossdating, but not for outbreak reconstruction. Therefore, usage of these methods cannot influence on correspondence of reconstruction results.

Really, we use different detrending methods in different detection algorithms (or don't use it at all). But in every case we just follow author's protocol. For OUTBREAK algorithm, we used detrending method (cubic spline) proposed by Swetnam and colleagues (Swetnam et al. USDA Forest Service PNW-RP-484, 1995). For independent components analysis the series have been normalized (with mean = 0 and sd = 1) and for the next step raw measurements have been detrended by cubic spline (Humbert, Kneeshaw, Forestry, 2011: 84(4)). For moving window method, the detrending procedure was excluded (Büntgen et al. New Phythologist, 2009: 182). Becker's algorithm (Becker et al. Ann Sci For, 1994: 51), the base of pointer year method, does not involve the usage of detrending too. Of course, we can't modify these protocols without enough cause.

L227 The object of the sentence is missing or you should remove “to”.

&

L227-231 Could you please clarify this description.

This paragraph have been completely rephrased (L228–232).

L245 Does independent component fast analysis correspond to MICA?

This method is the keystone for MICA, but not the same. In order to avoid this uncertainty and for better understanding, we rewrite the part of section concerning to MICA (L234–252).

L263 Consider to replace “weather” with “climate” as temperature and precipitation are generally considered climate data.

It is done (L275).

L276 It’s not clear how the disturbance factor has been included into the model. What do you mean for a dummy variable? Do you mean categorical?

Yes, it is. Dummy (or indicator) variable is a kind of categorical variable and it shows presence/absence of some feature. In the simplest case (as in this manuscript), dummy variables take the values 1 or 0. As far as I know, usage of dummy variables is most popular in econometrics, but it is known to ecologists too (e.g., Fu et al., J. of Forest Sciense, 2012: 58(3); Smale et al., Agricultural Economics, 2003: 28).

L276 How did you account for intraspecific variability of responses? E.g. using random factor in Linear mixed effect (LME) models? You could disentangle between the climate and outbreak effect considering both numerical (climate) and categorical (outbreak) variable in the models.

The detailed investigation of climate vs. outbreak effect on radial growth goes far beyond our goals. It is enough demonstrate the statistically significant increasing of model accuracy after defoliation effect incorporation. I guess, it has been reached (section 4.3 and Table 5) by dummy variables model. Besides, even if mixed effect models have got some advantages (Bell et al., Quality & Quantity, 2019: 53) in theory, but in real cases dummy variables models are quite helpful too: 'In terms of the three fit statistics …, there was no large difference between mixed-effects model and dummy variable model' (Fu et al., J. of Forest Sciense, 2012: 58(3)).

However, I'm very grateful to You for this idea. We will use mixed models necessarily in the further work dedicated to defoliation-climate-radial growth relationship.

L276 In addition, it is not clear if EW and LW width have been considered separately or not.

It was mentioned directly: 'linear models of relationship between tree-ring width [not early- or latewood] and climate conditions' (L292).

L287 after defoliation/crown reduction.

The sentence You mentioned have been removed from the text.

L323 Replace “Basing” with “based”.

It's done (L313).

L329 Please link to table or figure with the results. What do “optimal parameter values” mean?

Such clarify have been added: '(in terms of precise prediction of well-documented 1996–1999 and 2003–2008 outbreaks)' (L317–318). See also 3.4.5 section.

L404 Is the correlation significant?

&

L406 Radial growth also correlate with… did you also test EW and LW separately?

Thank You very much. We had not performed correlation analysis, and usage the term 'correlation' is incorrect. We have made a clarification which taking into account the statistical method has been used: '…there is a statistically significant at 0.05 level relationship (in terms of response analysis) between the ring width…' (L393–394). Besides, we have took into account Your comment about L406 (see citation above).

The discussion should take into account the results from the LME models.

The climate-growth relationship examination is just a by-product. I guess, for our goals it is quite enough the brief mention in 5.5.1 subsection. Absolutely, this relationship is very interesting matter, but if we will discuss it in details, than the text volume will became excessively huge.

L555 Here climate (and not weather as before) is mentioned. Is this a subchapter?

&

L573, 592, 596 Is it another subchapter?

Yes, they are. I have formatted these lines as subchapters.

L555 “The 1st possible reason” of what?

Thank You very much for this useful comment. We have revised subchapters 5.3.1, 5.3.3 and 5.3.4 in various degrees to avoid such drawbacks.

At last, we have tried to correct the quality of language. If the text still need proofreading and corrections we intend to use MDPI English Editing Service. 

Best wishes,

Denis Demidko and colleagues

Round 3

Reviewer 4 Report

The current version of the manuscript entitled “Pine looper Bupalus piniaria (L.) outbreaks reconstruction: a case study for Southern Siberia” ID: 996673 has again improved compared to the previous version.

The text would benefit from an English proofreading.

In addition please consider the specific minor comments below.

Abstract

L 34 Please replace “growth” with “width” as the reader could refer to primary or secondary growth.

L37 here you could just refer to linear model results.

L40 What do you mean for “official data”… “official records?”

L43 ..more detailed?..more precise? What do you mean with “better”?

L44 do you think this method is applicable only to Scots Pine or could be used with other species too?

Introduction

L60 Please clarify why the phyllophagus insects populations are economically important. I probably wrongly thought the Pinus sylvestris trees could be of economic importance for the people living in that territories.

L71 I guess the reference “Denis et Schiffermüller” is wrongly cited.

L106 Do you mean “carbohydrates produced during the photosynthesis”?

L124-126 Shouldn’t this part be in the discussion?

Methods

L172 Please replace “in” with “using”.

L175 ..”by cross dating the samples” I guess.

L178 Please check the verb forms through the manuscript. “Please replace “was” with “were”.

L196 If the relative growth change has been calculated for early and latewood you could consider to show both formulas to improve clarity.

L280 If p>=0.05 the selected characteristics were simply not significant.

L281 Instead of “naked eye” I suggest to use “observations”.

Discussion

L486 Please remove this sentence.

Author Response

Thank You very much for Your comments. Here we submit our corrections:

L34 Please replace “growth” with “width” as the reader could refer to primary or secondary growth.

The 'growth' have been replaced with 'width' (L34).

L37 here you could just refer to linear model results.

The sentence have been shortened to 'Linear modeling showed a difference between the climate impact on radial growth and the defoliation one' (L36–38). I hope I understand Your correctly. 

L40 What do you mean for “official data”… “official records?”

This word combination have been replaced with 'Forest Service' (L40).

L43 ..more detailed?..more precise? What do you mean with “better”?

As You recommend, we have used 'more precise' phrase (L44).

L44 do you think this method is applicable only to Scots Pine or could be used with other species too?

We assume this method can be useful for coniferous and ring-porous tree species. The sentence in L44–46 has been corrected according to this opinion.

L60 Please clarify why the phyllophagus insects populations are economically important. I probably wrongly thought the Pinus sylvestris trees could be of economic importance for the people living in that territories.

The remark about economic importance of pine looper damage has been added to L60–62.

L71 I guess the reference “Denis et Schiffermüller” is wrongly cited.

In Fauna Europaea (https://fauna-eu.org) and in Wang et al. PLoS ONE, 2014 9(3):e90598 this species noted as Panolis flammea (Denis & Schiffermüller, 1775). In the manuscript we have changed 'et' to ampersand (L74).

L106 Do you mean “carbohydrates produced during the photosynthesis”?

Thank You very much. The sentence has been corrected to 'Latewood is composed mainly of carbohydrates producing during the current year photosynthesis' (L109–110).

L124-126 Shouldn’t this part be in the discussion?

On the one hand, it would be right. On the other hand, however, the absence of this additional goal could be confusing. Readers can be perplexed if this issue will be absent in Introduction but will appeared in Discussion.

L172 Please replace “in” with “using”.

Thank You; the 'in' have been replaced (L178).

L175 ..”by cross dating the samples” I guess.

The sentence has been corrected according to Your proposal (L180).

L178 Please check the verb forms through the manuscript. “Please replace “was” with “were”.

Thank You very much. The 'was' have been replaced by 'were' in L177, 183, 246, 247.

L196 If the relative growth change has been calculated for early and latewood you could consider to show both formulas to improve clarity.

It should be noted that relative growth change has been used for ring width, too (3.4.2 section). Therefore, if we put three exactly the same equations in the manuscript it will lead to unreasonable increasing in the amount of text. I propose the compromise solution using indices in the equation (see Eq. 1 after L203, also L201–202).

L280 If p>=0.05 the selected characteristics were simply not significant.

Here we emphasize that we used 0.05 (but not 0.01, 0.001 or another) significance level. For better clarity, we have changed this part of sentence to 'demonstrate a statistically significant  response at 0.05 level' (L286).

L281 Instead of “naked eye” I suggest to use “observations”.

Thank You. We have made change according to Your proposal (L287).

L486 Please remove this sentence.

The sentence in this line has been removed (L494).

With best regards, 

Denis Demidko and colleagues. 
